# On Learning Necessary and Sufficient Causal Graphs

**Hengrui Cai**
University of California, Irvine
hengrc1@uci.edu

**Yixin Wang**
University of Michigan
yixinw@umich.edu

**Michael I. Jordan**
University of California, Berkeley
jordan@cs.berkeley.edu

**Rui Song**
North Carolina State University
songray@gmail.com

## Abstract

The causal revolution has stimulated interest in understanding complex relationships in various fields. Most of the existing methods aim to discover causal relationships among all variables within a complex large-scale graph. However, in practice, only a small subset of variables in the graph are relevant to the outcomes of interest. Consequently, causal estimation with the full causal graph—particularly given limited data—could lead to numerous *falsely discovered, spurious* variables that exhibit high correlation with, but exert no causal impact on, the target outcome. In this paper, we propose learning a class of *necessary and sufficient causal graphs (NSCG)* that exclusively comprises causally relevant variables for an outcome of interest, which we term *causal features*. The key idea is to employ *probabilities of causation* to systematically evaluate the importance of features in the causal graph, allowing us to identify a subgraph relevant to the outcome of interest. To learn NSCG from data, we develop a *necessary and sufficient causal structural learning (NSCSL)* algorithm, by establishing theoretical properties and relationships between probabilities of causation and natural causal effects of features. Across empirical studies of simulated and real data, we demonstrate that NSCSL outperforms existing algorithms and can reveal crucial yeast genes for target heritable traits of interest.

## 1   Introduction

Causal discovery has gained significant attention in recent years for disentangling complex causal relationships in various fields. Building upon the causal graphical model [see e.g., 23], many causal structural learning algorithms have been developed [see e.g., 35; 7; 34; 14; 4; 27; 46; 44; 48; 5] to infer the causal knowledge (e.g., causal graphs) from observed data. These algorithms are based on the assumption of causal sufficiency (the absence of unmeasured confounders). In real-world applications, to satisfy such an assumption, we strive to learn large-scale causal graphs [see e.g., 20; 6; 38; 21], in the hope of  **sufficiently** describing how an outcome of interest depends on its relevant variables.

In addition to sufficiency, it is also crucial to account for the concept of **necessity** by excluding redundant variables in explaining the outcome of interest. Failure to do so can result in the inclusion of spurious variables in the learned causal graphs, which are highly correlated but have no causal impact on the outcome. These variables can impede causal estimation with limited data and lead to falsely discovered spurious relationships, leading to poor generalization performance for downstream prediction [31]. For example, it might be observed that men aged 30 to 40 who buy diapers are also likely to buy beer. However, beer purchase is a spurious feature for diaper purchases: their correlation is not necessarily causal, as both purchases might be confounded by a shared cause, such as new

37th Conference on Neural Information Processing Systems (NeurIPS 2023).

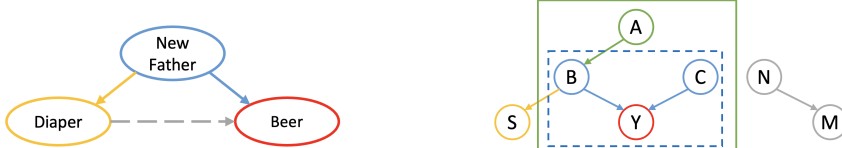

Figure 1: **Left**: Illustration of the causal relationship between the customer being a new father or not, beer purchasing, and diaper purchasing, where solid lines represent the true model, and the dashed line corresponds to the spurious correlation between beer purchasing and diaper purchasing. **Right**: Relationship between various causal structures. The nodes $A$, $B$, and $C$ belong to the necessary and sufficient causal graph for the desired result $Y$ and are represented within the solid green square. Among them, nodes $B$ and $C$ are members of the Markov blanket of $Y$, enclosed by the blue square. The node $S$ is the spurious variable for $Y$, while the nodes $N$ and $M$ are not related to the target.

fathers buying diapers for childcare while also buying beer to alleviate stress. Therefore, simply increasing the availability of diapers or beer will not causally improve the demand for the other (see also Fig. 1(left)).

Furthermore, the number of variables causally relevant to the outcome of interest is often considerably smaller than the number of variables included in estimating a causal graph (see Fig. 1(right)). For example, while an individual's genome may encompass 4 to 5 million single nucleotide polymorphisms (SNPs), only a limited number of non-spurious genes or proteins are found to systematically regulate the expression of the phenotype of interest [e.g., 6]. Similarly, in natural language processing tasks, excluding spurious embeddings such as writing style and dialect can enhance model accuracy and downstream prediction performance [e.g., 10]. Thus, a more parsimonious causal graph is required to unveil the necessary and sufficient causal dependencies.

In this work, we focus on learning *necessary and sufficient causal graphs* (NSCG) that only contain causally relevant variables (which we term *causal features*) for an outcome of interest, offering a compact representation of causal graphs for a target outcome. Our **contributions** are three-fold.

• We propose the notion of NSCG (see an illustration inside the green solid square in the right panel of Fig. 1). The key idea is to leverage the marginal and conditional probabilities of causation (POC) to systematically characterize the importance of variables (a.k.a. features).
• We establish theoretical properties and relationships between POC and the natural causal effects of features, and derive the conditions under which they are equivalent, with lower bounds provided for identification. The natural causal effects of features have explicit forms under parametric models such as the linear structural equation model, enabling convenient estimation of the POC from observed data.
• To select necessary and sufficient features in causal graphs, we propose a necessary and sufficient causal structural learning algorithm (NSCSL) to learn an NSCG containing all necessary and sufficient causes without unnecessary spurious features. This enables feature selection for causal discovery.

The proposed method provides concise explanations of causal relationships with high-dimensional data (i.e., with a large number of variables). Empirical studies in simulated datasets show that NSCSL outperforms existing algorithms in distilling relevant subgraphs for outcomes of interest; NSCSL can also identify important quantitative trait loci for the yeast and the causal protein signaling network for single cell data, as demonstrated in real data analyses.

## 1.1 Related Works

The literature on *causal structural learning* can be broadly classified into three classes. The first class of methods focuses on using local conditional independence tests to identify the causal skeleton and determine the direction of the edges, such as the PC algorithm [35; 14; 36]. The second class of methods uses functional causal models with additional assumptions about the data distribution, including ICA-LiNGAM [34] and the causal additive model (CAM) [4]. The third class, the score-based methods, includes greedy equivalence search (GES) [7; 27; 11] and acyclicity optimization methods [46]. Refer to [44; 48; 17; 5; 47; 41] for additional cutting-edge causal structural learning methods. Yet, these works do not consider the necessity of the variables incorporated in the causal graph, i.e., whether the variables are causally relevant to the outcome. Consequently, such algorithms can often produce a redundant or potentially misleading graph, as depicted in Fig. 1 (right).

Our work also links to *feature selections* [see an overview in 16]. Despite the extensive literature, only a few studies have examined variable selection in causal graphs. One notable exception is Aliferis et al. [1], which uses the concept of the Markov blanket to construct a local causal graph for the target variable of interest. In this context, a Markov blanket of a variable $Y$ is the minimal variable subset conditioned upon which all other variables become probabilistically independent of $Y$. Consequently, their algorithm uncovers only direct parents or children in the identified causal graph (such as the blue dotted square in the right panel of Fig. 1) and thereby overlooks the ancestors that contain atavistic information and indirectly influence the outcome. Recent works [18; 19] consider a minimal sufficient action set in bandits. Yet, these methods [also see 12; 13] rely on a true or known graph. We instead propose to simultaneously learn the causal graph and select the causal features.

Lastly, our work is connected to the body of research on *probability of causation* [e.g., 22; 39; 45], which delineates the necessity and sufficiency of features for the outcome of interest. Recently, Wang & Jordan [42] introduced this concept into representation learning, formulating the non-spuriousness and efficiency of representations by generalizing the probabilities of causation to accommodate low-dimensional representations of high-dimensional data. However, these works primarily concentrate on the identification of probabilities of causation, assuming that the causal graph among the variables under consideration is known with causally independent features. We address this gap in our work by incorporating the notion of probabilities of causation into learning complex causal graphs.

## 2 Framework

**Graph terminology.** Consider a graph $\mathcal{G} = (\boldsymbol{X}, \boldsymbol{D_X})$ with a node set $\boldsymbol{X}$ and an edge set $\boldsymbol{D_X}$ that encompasses all edges in $\mathcal{G}$ for nodes $\boldsymbol{X}$. A node $X_i$ is said to be a parent of $X_j$ if there is a directed edge from $X_i$ to $X_j$, i.e., $X_i$ is a direct cause of $X_j$. A node $X_k$ is said to be an ancestor of $X_j$ if there is a directed path from $X_k$ to $X_j$ regulated by at least one additional node $X_i$ for $i \neq k$ and $i \neq j$, i.e., $X_k$ is an indirect cause of $X_j$. Let the set of all parents/ancestors of node $X_j$ in $\mathcal{G}$ as $\text{PA}_{X_j}(\mathcal{G})$. A directed graph $\mathcal{G}$ that does not contain directed cycles is called a directed acyclic graph (DAG). The structural causal model (SCM) characterizes the causal relationship among $|\boldsymbol{X}| = d$ nodes via a DAG $\mathcal{G}$ and noises $\boldsymbol{e_X} = [e_{X_1}, \cdots, e_{X_d}]^\top$ such that $X_i := h_i\{\text{PA}_{X_i}(\mathcal{G}), e_{X_i}\}$ for some unknown $h_i$ and $i = 1, \cdots, d$.

**Notations and assumptions.** Denote $\boldsymbol{O} = (\boldsymbol{Z}, Y)$ as a collection of nodes that contains features $\boldsymbol{Z} = [Z_1, \cdots, Z_p]^\top \in \mathcal{Z} \subset \mathbb{R}^p$ and a discrete outcome of interest as $Y \in \mathcal{L} = \{y_1, \cdots, y_l\}$ for $l$ different values. Here, the features can be intervened, such as treatment and mediators. Let $Y(\boldsymbol{Z} = \boldsymbol{z})$ be the potential value of $Y$ that would be observed after setting variable $\boldsymbol{Z}$ as $\boldsymbol{z}$. This is equivalent to the value of $Y$ by imposing a 'do-operator' of $do(\boldsymbol{Z} = \boldsymbol{z})$ as in Pearl et al. [23]. Similarly, one can define the potential outcome, $Y(Z_i = z_i)$, by setting an individual variable $Z_i$ as $z_i$, while keeping the rest of the model unchanged. Suppose there exists an SCM that characterizes the causal relationship among $\boldsymbol{O}$, with its DAG as $\mathcal{G}_{\boldsymbol{O}}$. A notation and abbreviation table is provided in App. A. Following the causal inference literature [see e.g., 29; 22; 23; 42], we assume:

(A1). **Consistency**: $\boldsymbol{Z} = \boldsymbol{z} \leftrightarrow Y(\boldsymbol{Z} = \boldsymbol{z}) = Y, \forall \boldsymbol{z} \in \mathcal{Z}$.
(A2). **Ignorability**: (i) $Y(\boldsymbol{Z} = \boldsymbol{z}) \perp \boldsymbol{Z}, \forall \boldsymbol{z} \in \mathcal{Z}$;    (ii) $Y(Z_i = z_i) \perp Z_i | \text{PA}_{Z_i \cup Y}(\mathcal{G}_{\boldsymbol{O}}), \forall z_i \in \mathcal{Z}_i$.

Here, (A1) implies that the outcome observed for each unit under study with features as $\boldsymbol{z}$ is identical to the outcome we would have observed had that unit been set with features $\boldsymbol{Z} = \boldsymbol{z}$. In addition, since we include as many confounders as possible, the ignorability assumption in (A2), also known as the no unmeasured confounderness assumption, is satisfied.

## 3 Necessary and Sufficient Causal Graphs

We care about a subset or a function of $\boldsymbol{Z}$, denoted as $\boldsymbol{X} = [X_1, \cdots, X_d]^\top$ (of $d$ dimension with possibly $d \ll p$), which indeed captures the causal relationship between $\boldsymbol{Z}$ and $Y$. To be specific, let an SCM for causal nodes $\boldsymbol{V} = (\boldsymbol{X}, Y)$ with its DAG as $\mathcal{G}_{\boldsymbol{V}} = (\boldsymbol{V}, \boldsymbol{D_V})$ and $e_{\boldsymbol{V}}$ as a $d+1$ dimensional independent noise, to characterize the causal relationship between $\boldsymbol{X}$ and $Y$. Let $\mathbb{P}_{\mathcal{G}}$ be the mass/density function for an SCM with its DAG $\mathcal{G}$. Following the causal (or disentangled) factorization in the causal graphical model [23], we define the *sufficient* causal graph as follows.

**Definition 3.1.** (Sufficient Graph) The graph $\mathcal{G}_{\boldsymbol{V}}$ is a *sufficient* causal graph to capture the causal relationship among $\boldsymbol{Z}$ and $Y$ with $\boldsymbol{X} \subset \boldsymbol{Z}$ or $\boldsymbol{X} = f(\boldsymbol{Z})$ (where $f$ is within a countable or Vapnik-Chervonenkis (VC) class) if $\mathbb{P}_{\mathcal{G}_{\boldsymbol{V}}}\{Y | \text{PA}_Y(\mathcal{G}_{\boldsymbol{V}})\} \prod_{X_i \in \text{PA}_Y(\mathcal{G}_{\boldsymbol{V}})} \mathbb{P}_{\mathcal{G}_{\boldsymbol{V}}}\{X_i | \text{PA}_{X_i}(\mathcal{G}_{\boldsymbol{V}})\}$ $= \mathbb{P}_{\mathcal{G}_{\boldsymbol{O}}}\{Y | \text{PA}_Y(\mathcal{G}_{\boldsymbol{O}})\} \prod_{Z_i \in \text{PA}_Y(\mathcal{G}_{\boldsymbol{O}})} \mathbb{P}_{\mathcal{G}_{\boldsymbol{O}}}\{Z_i | \text{PA}_{Z_i}(\mathcal{G}_{\boldsymbol{O}})\}$.

Here, Def. 3.1 refers to a sub-structure $\mathcal{G}_{\boldsymbol{V}}$ (from the whole graph $\mathcal{G}_{\boldsymbol{O}}$) containing all directed edges or paths towards $Y$, making it sufficient to describe how $Y$ depends on all its ancestors. Then, the causal graph $\mathcal{G}_{\boldsymbol{V}}$ is said to be *necessary* and *sufficient* if it satisfies the following definition.

**Definition 3.2.** (Necessary and Sufficient Graph) Suppose $\mathcal{G}_{\boldsymbol{V}}$ satisfies Def. 3.1, then $\mathcal{G}_{\boldsymbol{V}}$ is a *necessary* and *sufficient* causal graph to capture the causal relationship among $\boldsymbol{Z}$ and $Y$ if for any true subset $\boldsymbol{W}$ of $\boldsymbol{X}$, i.e., $\boldsymbol{W} \subset \boldsymbol{X}$ or $\boldsymbol{W} = g(\boldsymbol{X})$ (where $g$ is within a countable or VC class), with $\boldsymbol{U} = (\boldsymbol{W}, Y)$, we have $\mathbb{P}_{\mathcal{G}_{\boldsymbol{V}}}\{Y|\text{PA}_Y(\mathcal{G}_{\boldsymbol{V}})\} \prod_{X_i \in \text{PA}_Y(\mathcal{G}_{\boldsymbol{V}})} \mathbb{P}_{\mathcal{G}_{\boldsymbol{V}}}\{X_i|\text{PA}_{X_i}(\mathcal{G}_{\boldsymbol{V}})\} \neq \mathbb{P}_{\mathcal{G}_{\boldsymbol{U}}}\{Y|\text{PA}_Y(\mathcal{G}_{\boldsymbol{U}})\} \prod_{W_i \in \text{PA}_Y(\mathcal{G}_{\boldsymbol{U}})} \mathbb{P}_{\mathcal{G}_{\boldsymbol{U}}}\{W_i|\text{PA}_{W_i}(\mathcal{G}_{\boldsymbol{U}})\}$, where $\mathcal{G}_{\boldsymbol{U}}$ is the causal graph for $\boldsymbol{U}$.

Therefore, by Def. 3.2, we can further identify the *minimal sub-structure* $\mathcal{G}_{\boldsymbol{V}}$ which includes only all directed edges or paths leading to $Y$. The goal is to learn such a necessary and sufficient causal graph (NSCG) $\mathcal{G}_{\boldsymbol{V}}$ from the observed data denoted as $\{\boldsymbol{o}^{(j)} = (\boldsymbol{z}^{(j)}, y^{(j)})\}_{1 \leq j \leq n}$ with sample size $n$, by identifying the latent causal features $\boldsymbol{X}$. Denote the resulting estimated graph as $\widehat{\mathcal{G}}_{\boldsymbol{V}}$.

# 4 Probability of Causation and Causal Effects

Obtaining an NSCG $\mathcal{G}_{\boldsymbol{V}}$ directly based on Def. 3.2 poses several challenges, as the latent causal features $\boldsymbol{X}$ driving the causal graph remain unknown. A naïve approach is to search all different combinations of $\boldsymbol{Z}$ for a candidate of $\boldsymbol{X}$ such that Def. 3.2 holds, which yields a complexity of $\mathcal{O}(p^p)$. This motivates us to assess the necessity and sufficiency of features in determining the outcome by introducing the concepts of probabilities of causation and causal effects, which will be elaborated on and interconnected in this section.

## 4.1 Probabilities of Causation and Lower Bounds

The probabilities of a feature being necessary and sufficient, known as the probability of causation (POC), have been proposed and studied [see 22; 39; 42]. Specifically, the probability of necessity and sufficiency (PNS) of feature $\boldsymbol{Z}$ is first defined in Tian & Pearl [39] as follows.

**Definition 4.1.** PNS in Tian & Pearl [39] with a univariate binary feature $Z$:

$$PNS \equiv \mathbb{P}\{Y(Z \neq z) \neq y, Y(Z = z) = y\} \underset{by\,(A1)}{=} \mathbb{P}(Z = z, Y = y)\cdot PN + \mathbb{P}(Z \neq z, Y \neq y)\cdot PS,$$

where the probability of necessity (PN) is $PN = \mathbb{P}\{Y(Z \neq z) \neq y|Z = z, Y = y\}$, and the probability of sufficiency (PS) is $PS = \mathbb{P}\{Y(Z = z) = y|Z \neq z, Y \neq y\}$.

The second equation in Def. 4.1 holds under (A1) [see details in 22; 39]. The PN score reflects the necessity of $Z$ by evaluating the probability of the outcome becoming worse if revising the features given the good outcome observed. Similarly, the PS score indicates the sufficiency of $Z$ by evaluating the probability of the outcome becoming better if changing the features given the bad outcome observed. Therefore, the PNS score shows the causal importance of the features by combining necessary and sufficient properties. The above definition can be generalized to multivariate cases for nonbinary features [see e.g., 42] to quantify the POC of an individual feature $Z_i$. Let $\boldsymbol{Z}_{-i} \equiv \boldsymbol{Z} \setminus Z_i$ be the set of complementary variables of $Z_i$. In the following, we consider two different POCs for $Z_i$ by extending the work of Wang & Jordan [42].

**Definition 4.2.** Marginal POC (M-POC) for $Z_i$:

$$\text{M-POC}_i(y) \equiv \mathbb{P}\{Y(Z_i \neq z_i) \neq y, Y(Z_i = z_i) = y\}.$$

**Definition 4.3.** Conditional POC (C-POC) for $Z_i$:

$$\text{C-POC}_i(y) \equiv \mathbb{P}\{Y(Z_i \neq z_i, \boldsymbol{Z}_{-i} = \boldsymbol{z}_{-i}) \neq y, Y(Z_i = z_i, \boldsymbol{Z}_{-i} = \boldsymbol{z}_{-i}) = y\}.$$

We introduce the marginal POC (M-POC) as a novel quantity in the literature to summarize the overall causal importance of an individual feature in determining the outcome's value. The conditional POC (C-POC) in Def. 4.3 corresponds to the conditional PNS in Wang & Jordan [42], which quantifies the likelihood of an individual feature being a direct and significant cause of the outcome while holding other features constant. As per Section 9.2.3 in Pearl et al. [22], the PNS in Def. 4.1 is not estimable unless additional conditions (monotonicity) are specified. To alleviate such a condition for identifying Defs. 4.2 and 4.3, we derive the lower bounds for the proposed POCs as follows.

**Theorem 4.4.** *(Lower Bound of Probabilities of Causation) Suppose (A1) and (A2) hold. Then*

$$M\text{-}POC_i(y) \geq \mathbb{P}(Y = y|Z_i = z_i) - \mathbb{P}(Y = y|Z_i \neq z_i),$$
$$C\text{-}POC_i(y) \geq \mathbb{P}(Y = y|Z_i = z_i, \boldsymbol{Z}_{-i} = \boldsymbol{z}_{-i}) - \mathbb{P}(Y = y|Z_i \neq z_i, \boldsymbol{Z}_{-i} = \boldsymbol{z}_{-i}).$$

Table 1: Causal effects from the customer being a new father or not ($X_F$) and diaper purchasing ($X_D$) on beer purchasing ($X_B$), where $\omega_1, \omega_2 \in (0, 1)$ with $\omega_1 + \omega_2 = 1$ and $c \to 1$, in the estimated causal graph ($X_F \xrightarrow{\omega_1} X_B$, and $X_F \xrightarrow{c} X_D \xrightarrow{\omega_2} X_B$).

| Variable Name | Direct Effect | Total Effect |
|---|---|---|
| New Father ($X_F$) | $\omega_1$ | $\omega_1 + c\omega_2 \to 1 \quad (> \omega_2)$ |
| Diaper ($X_D$) | $\omega_2$ | $\omega_2$ |

The proofs of Thm. 4.4 are in App. D. The lower bound equality holds when an additional monotonicity condition is imposed, with details in App. D. The results in Thm. 4.4 allow us to estimate the lower bound of POC from observed data by learning the conditional probability of $Y$ given various combinations of confounders. This, in turn, aids in evaluating the significance of features, with details provided in App. B. Yet, estimating these conditional probabilities of $Y$ based on high-dimensional features is very challenging [e.g., 32; 42], which motivates us to consider the corresponding expected mean outcome given different combinations of the confounders.

### 4.2 Causal Effects and Connection to POCs

To connect the proposed POCs and facilitate the empirical estimation, we introduce the natural total effect (TE) and natural direct effect (DE) for $Z_i$ by extending definitions in Pearl et al. [22].

**Definition 4.5.** Natural Causal Effects for $Z_i$:

$$TE_i = \mathbb{E}\{Y(Z_i = z_i + 1)\} - \mathbb{E}\{Y(Z_i = z_i)\},$$
$$DE_i = \mathbb{E}\{Y(Z_i = z_i + 1, \boldsymbol{Z}_{-i} = \boldsymbol{z}_{-i}^{(z_i)})\} - \mathbb{E}\{Y(Z_i = z_i)\},$$

where $\boldsymbol{z}_{-i}^{(z_i)}$ is the value of $\boldsymbol{Z}_{-i}$ if setting $do(Z_i = z_i)$.

The natural total effect ($TE_i$) can be understood as the marginal change in the outcome when increasing $Z_i$ by one unit. Similarly, the natural direct effect ($DE_i$) represents the conditional change in the outcome when $Z_i$ is increased by one unit, with all other features held constant. Indeed, the natural total and direct causal effects delineated in Pearl et al. [22] emerge as particular instances of Def. 4.5 when examining a solitary treatment subject to intervention. By comparing Def. 4.5 with Defs. 4.2 and 4.3, it is natural to establish the relationship between POCs and causal effects below.

**Theorem 4.6.** *(Relation between POCs and Causal Effects) Define* $\delta_M(z_i) \equiv \mathbb{E}\{Y|Z_i = z_i\} - \mathbb{E}\{Y|Z_i \neq z_i\}$ *and* $\delta_C(z_i) \equiv \mathbb{E}\{Y|Z_i = z_i, \boldsymbol{Z}_{-i} = \boldsymbol{z}_{-i}\} - \mathbb{E}\{Y|Z_i \neq z_i, \boldsymbol{Z}_{-i} = \boldsymbol{z}_{-i}\}$. *Suppose* *(A1)-(A2) hold, then*

$$\sum_{y \in \mathcal{L}} y M\text{-}POC_i(y) \geq \delta_M(z_i), \qquad \sum_{y \in \mathcal{L}} y C\text{-}POC_i(y) \geq \delta_C(z_i),$$

*if* $Y$ *is nonnegative. Further, if* $Z_i$ *is binary, we have*

$$\min\{\sum_{y \in \mathcal{L}} y M\text{-}POC_i(y), |TE_i|\} \geq \delta_M(z_i), \qquad \min\{\sum_{y \in \mathcal{L}} y C\text{-}POC_i(y), |DE_i|\} \geq \delta_C(z_i).$$

The proofs of Thm. 4.6 can be found in App. D, with the lower bound equality holds when an additional monotonicity condition is imposed. We can summarize the findings of Thms. 4.4 and 4.6 in two key aspects. First, the marginal and conditional POCs assess the *likelihood* of a feature being spurious, while the absolute values of natural causal effects quantify the *size* of such a spurious effect based on the magnitude of the outcome of interest. Both are lower bounded by the same quantity, that is, the differences in expectations based on the corresponding POC, given non-negative outcomes and binary features. With appropriate data processing, we can transform the outcome to be nonnegative, and thus causal effects become a suitable substitute for POCs. Second, these two approaches exhibit consistency under the monotonicity condition. The natural causal effects of features have explicit forms under parametric models (see details in § 5.1), such as the linear structural equation model, enabling convenient estimation of the necessity and sufficiency of features from observed data. This section concludes with a toy example illustrating the identification of spurious features through the proposed causal effects and the distinction between total and direct causal effects.

**Example 4.7.** *(Recall: Beer and Diaper) Given the causal graph in Fig. 1(left), consider a linear SCM for the customer being a new father or not ($X_F$), diaper purchasing ($X_D$), and beer purchasing ($X_B$): $X_D = X_F + e_D$ and $X_B = X_F + e_B$, where $e_D$ and $e_B$ are independent mean zero noises.*

*Since the true SCM is unknown, fitting a linear model $X_B \sim \omega_1 X_F + \omega_2 X_D$ would result in ambiguous coefficients $\omega_1, \omega_2 \in [0,1]$ with $\omega_1 + \omega_2 = 1$. By fitting $X_D \sim cX_F$ and obtaining $c$ close to 1, we have the estimated causal graph as $X_F \xrightarrow{\omega_1} X_B$, and $X_F \xrightarrow{c} X_D \xrightarrow{\omega_2} X_B$. Based on Def. 4.5, we estimate the corresponding causal effects in Table 1. It can be observed that the total effect of $X_F$ on $X_B$ surpasses the total effect from $X_D$, indicating the spurious nature of the diaper purchasing, which should be removed to form the desired NSCG. Yet, using the direct effect may not be able to distinguish their differences due to the high correlation between $X_F$ and $X_D$ if $\omega_1 < \omega_2$.*

## 5    Necessary and Sufficient Causal Structural Learning

In this section, we formally present how to learn NSCG. Based on Thms. 4.4 and 4.6, a simple solution is to first use a pre-screening process to find necessary and sufficient features from $\boldsymbol{Z}$ that achieve high scores of causation, and then estimate the causal graph among the selected nodes and $Y$ to approximate $\mathcal{G}_{\boldsymbol{V}}$. This approach works for general SCMs while may suffer from overfitting. Instead of using such a two-step learning, we propose to learn necessary and sufficient features and the causal graph simultaneously through a single-step optimization. To this end, in § 5.1, we first introduce the structural equation model in order to provide the closed-form expressions of the proposed causal quantities. The main algorithm based on causal effects is presented in § 5.2 for the *linear* model, with the POC-based version available in App. B for the *nonlinear* model.

### 5.1    Structural Equation Model and Close Form of Causal Effects

**Structural equation model.** We define a selection function $g$ that maps the feature set $\boldsymbol{Z}$ to a subset, aiming to maintain good interpretability. That is, $g : \boldsymbol{Z} \in \mathbb{R}^p \to g(\boldsymbol{Z}) \in \mathbb{R}^d$ where $d \ll p$, and we denote the $i$-th dimension of $g(\boldsymbol{Z})$ as $g_i(\boldsymbol{Z})$. Following the causal structure learning literature [35; 25; 46; 44; 48; 5], we assume the Markov and faithfulness conditions and consider a linear structural equation model (LSEM) such that $\{g(\boldsymbol{Z}), Y\}$ is characterized by the pair $(\boldsymbol{B}, \boldsymbol{\epsilon})$ as

$$\begin{bmatrix} g(\boldsymbol{Z}) \\ Y \end{bmatrix} \leftarrow \boldsymbol{B} \begin{bmatrix} g(\boldsymbol{Z}) \\ Y \end{bmatrix} + \boldsymbol{\epsilon} \equiv \begin{bmatrix} \boldsymbol{B}_g & 0 \\ \boldsymbol{\theta} & 0 \end{bmatrix} \begin{bmatrix} g(\boldsymbol{Z}) \\ Y \end{bmatrix} + \begin{bmatrix} \boldsymbol{\epsilon}_{\boldsymbol{Z}} \\ \epsilon_Y \end{bmatrix}, \tag{1}$$

where $\boldsymbol{B}$ is a $(d+1) \times (d+1)$ weighted adjacent matrix that characterizes the causal relationship among $\{g(\boldsymbol{Z}), Y\}$, and $\boldsymbol{\epsilon} \equiv [\boldsymbol{\epsilon}_{\boldsymbol{Z}}^\top, \epsilon_Y]^\top$ is a $d + 1$ dimensional random vector of jointly independent errors. Here, $\boldsymbol{B}$ consists of three components: (1). a $d \times d$ matrix $\boldsymbol{B}_g = \{b_{i,j}\}_{1 \le i \le d, 1 \le j \le d}$ with $b_{i,j}$ as the weight of the edge $g_i(\boldsymbol{Z}_i) \to g_j(\boldsymbol{Z}_i)$ if exists and $b_{i,j} = 0$ otherwise; (2) a $1 \times d$ vector $\boldsymbol{\theta} = [\theta_1, \cdots, \theta_p]$ for $\theta_i$ presenting the weight of the direct edge $g_i(\boldsymbol{Z}) \to Y$; and (3). a $(d+1) \times 1$ zero vector indicating the outcome of interest $Y$ cannot be any parent of the features. Without further assumptions, the model in (1) given a particular selector $g$ can be identified only up to a Markov equivalence class (MEC) [35; 25]. In the following, we focus on cases where the DAG can be uniquely identifiable, such as LSEM with Gaussian noises of equal variance [35; 26; 24], and linear model with non-Gaussian noise [34; 47]. See more details and extensions to MEC in App. C.1.

**Close form and estimation of causal effects.** We next provide the close forms of the causal effects in Def. 4.5 under the model in (1). Recall that $\theta_i$ presents the weight of the direct edge $g(\boldsymbol{Z})_i \to Y$. According to (1) and Def. 4.5, we have

$$DE_i(\boldsymbol{B}; g) = \theta_i.$$

The total causal effect can be quantified by the path method [see e.g., 43; 20]. Specifically, the causal effect of $g_i(\boldsymbol{Z})$ on $g_j(\boldsymbol{Z})$ along a directed path from $g_i(\boldsymbol{Z}) \to g_j(\boldsymbol{Z})$ in $\mathcal{G}$ can be calculated by multiplying all edge weights along the path, under LSEM. Denote the set of directed paths that starts with $g_i(\boldsymbol{Z})$ and ends with $Y$ as $\pi_i = \{g_i(\boldsymbol{Z}) \to \cdots \to Y\}$ with the size as $m_i$. Then the causal effect of $g_i(\boldsymbol{Z})$ on $Y$ through the directed path $\pi_i^{(k)} = \{i, l_1, \cdots, l_{\tau_k}, d+1\} \in \pi_i$ with length $\tau_k + 1$ is $PE\{\pi_i^{(k)}\} = b_{i,l_1} \cdots b_{l_{\tau_k},(d+1)}$, by the path method, where $b_{i,j}$ is the weight of the edge $g_i(\boldsymbol{Z}) \to g_j(\boldsymbol{Z})$ if it exists, and $b_{i,j} = 0$ otherwise, for $i, j \in \{1, \cdots, d\}$, and $b_{l_{\tau_k},(d+1)} = \theta_{l_{\tau_k}}$ as the direct edge from $g_{l_{\tau_k}}(\boldsymbol{Z})$ to $Y$. Thus,

$$TE_i(\boldsymbol{B}; g) = \sum_{k=1}^{m_i} PE\{\pi_i^{(k)}\}.$$

Both $TE_i$ and $DE_i$ can be explicitly calculated given a matrix $\boldsymbol{B}$ under a selector $g$. We denote their estimates as $\widehat{TE}_i$ and $\widehat{DE}_i$ given the estimated matrix $\widehat{\boldsymbol{B}}$ and $g$.

## 5.2 Learning Algorithm based on Causal Effects

The primary algorithm based on causal effects comprises three steps that quantify two sources of loss and learn the causal graph, specifically: loss of causal structural learning, loss of discovering causal features, and minimizing the overall loss to learn NSCG based on data $\{o^{(j)} = (z^{(j)}, y^{(j)})\}_{1 \leq j \leq n}$.

**Step 1: Form the loss from causal structural learning.** To estimate the matrix $B$ in (1), we adopt the acyclicity constraint [44; 46] as $h_1(B) \equiv \text{tr}\big[(I_{d+1} + tB \circ B)^{d+1}\big] - (d+1) = 0$, where $I_{d+1}$ is a $d+1$-dimensional identity matrix, and $\text{tr}(\cdot)$ is the trace of a matrix and $t$ is a hyperparameter that depends on the estimated largest eigenvalue of $B$. The first loss by the augmented Lagrangian is

$$L_1(B, g, \theta, \lambda_1 | \{o^{(j)}\}) = f(B, g, \theta | \{o^{(j)}\}) + \lambda_1 h_1(B), \tag{2}$$

where $f(B, g, \theta | \{o^{(j)}\})$ is some loss such as the least square error in NOTEARS [46] or the Kullback-Leibler divergence in DAG-GNN [44] with parameters $\theta$, and $\lambda_1$ is the Lagrange multiplier. Other causal structural leaning algorithms [see e.g., 35; 7; 34; 14; 4; 27; 48] can also be applied by formulating the corresponding score or loss function.

**Step 2: Constraints for causal relevance and causal identifiability.** We next measure the causal relevance of the selection function $g$ by the natural causal effects. We convert the outcome $Y$ to be nonnegative. According to Thm. 4.6, we can avoid the estimation of POCs by using the related causal effects in Def. 4.5 with their explicit expressions. This part of loss thus becomes

$$L_2^{CE}(B, g, \gamma | \{o^{(j)}\}) = -\sum_{i=1}^{d} |\widehat{CE}_i(B; g)| + \sum_{i=1}^{d+1} |b_{i,d+1}| + \gamma|g|. \tag{3}$$

Here, $\widehat{CE}_i$ can either take the estimated $DE_i$ or the estimated $TE_i$ given the matrix $B$ and $g$, as detailed in § 5.1. The second term corresponds to the causal identification constraint on the last column of $B$, requiring all elements to be zeros as in (1). This constraint restricts the causal structural learning to a smaller class of DAGs. Finally, $|g|$ denotes the number of selected nodes in $g$, with a penalty $\gamma$ to control the complexity of the selector.

**Step 3: Necessary and sufficient causal structural learning.** Combining two sources of loss functions in (2) with (3), leads to the objective as

$$\min_{B, g} \left[ L_1(B, g, \theta, \lambda_1 | \{o^{(j)}\}) + \alpha L_2^{CE}(B, g, \gamma | \{o^{(j)}\}) \right], \tag{4}$$

where $\alpha$ can be reviewed as a trade-off parameter between two loss functions. Next, we provide a solution for (4) *without* tuning $\alpha$. Specifically, based on no unmeasured confounders in (A2), we can calculate the highest causal effects that could be achieved given all variables without any penalty. Denote the estimated highest absolute causal effects in data as $\delta^*$. The goal is to find a subset of $Z$ such that $g(Z)$ achieves a similar level of necessity and sufficiency, i.e., the resulting score is close to $\delta^*$. Hence, we set the second loss as a constraint by comparing the overall causal effects of the selected nodes with the highest reference over the entire observed feature space, i.e.,

$$h_2(B; g) = \delta^* - \sum_{i=1}^{d} |\widehat{CE}_i(B; g)| + \sum_{i=1}^{d+1} |b_{i,d+1}|,$$

should be approaching 0 given a good selector $g$. This yields a new objective function as

$$\min_{B, g} f(B, g, \theta | \{o^{(j)}\}) + \lambda_1 h_1(B) + \lambda_2 h_2(B; g) + c|h_1(B)|^2 + d|h_2(B; g)|^2 + \gamma|g|, \tag{5}$$

where $\lambda_2$ is the Lagrange multiplier for the new constraint, and $c$ and $d$ are penalty terms. To minimize the loss in (5) and satisfy both $h_1(B) \to 0$ and $h_2(B; g) \to 0$, we simultaneously update $\lambda_1$ and $\lambda_2$ and increase $c$ and $d$ to infinity, by modifying the updating technique [see e.g., 46; 44] for multiple constraints, with the class of functions $g$ specified as the subset of $Z$ and its penalty $\gamma$ as the size of selected nodes in $g$. Here, the minimization can be solved using a black-box stochastic optimization such as 'Adam' [15]. Denote the estimated matrix as $\widehat{B}$, based on which we can obtain the estimated causal graph as $\widehat{\mathcal{G}}_V$ consisting of nodes $\widehat{g}(Z)$. Finally, we name this proposed algorithm as necessary and sufficient causal structural learning (NSCSL). The computational complexity of NSCSL is provided in App. B.3. We next establish the consistency of estimated causal graphs below.

**Theorem 5.1.** *Assume Model* (1) *holds with independent Gaussian error and equal variance. Suppose the topological ordering of the true bounded matrix $B$ is consistently estimated. Then the estimated matrix $\widehat{B}$ minimizing the loss in* (5) *converges to $B$ with the probability going to 1 as $n \to \infty$.*

Table 2: Comparison results across S1 to S3 under different sample sizes ($n$). Methods are evaluated by FDR, TPR, and SHD, with the standard error (SE) reported for each metric, over 50 replications.

| Scenario | Method | FDR±SE | | TPR±SE | | SHD±SE | |
|---|---|---|---|---|---|---|---|
| | | $n_1$ (small) | $n_2$ (large) | $n_1$ (small) | $n_2$ (large) | $n_1$ (small) | $n_2$ (large) |
| S1 | NSCSL-TE | 0.09±0.03 | 0.02±0.01 | 0.95±0.02 | 1.00±0.00 | 0.64±0.20 | 0.14±0.06 |
| $p=5$ | NSCSL-DE | 0.08±0.03 | 0.02±0.01 | 0.95±0.03 | 1.00±0.00 | 0.72±0.20 | 0.14±0.06 |
| $n_1=30$ | NOTEARS | 0.39±0.01 | 0.34±0.00 | 0.96±0.02 | 1.00±0.00 | 2.56±0.14 | 2.02±0.01 |
| $n_2=100$ | PC | 0.53±0.02 | 0.53±0.01 | 0.47±0.05 | 0.41±0.01 | 3.12±0.11 | 3.20±0.07 |
| ER Model | LiNGAM | 0.31±0.02 | 0.33±0.01 | 0.78±0.02 | 0.98±0.01 | 2.30±0.10 | 2.00±0.00 |
| S2 | NSCSL-TE | 0.10±0.03 | 0.02±0.01 | 1.00±0.00 | 0.99±0.01 | 0.38±0.14 | 0.12±0.06 |
| $p=5$ | NSCSL-DE | 0.12±0.04 | 0.01±0.01 | 0.67±0.04 | 0.50±0.00 | 1.00±0.12 | 1.02±0.01 |
| $n_1=30$ | NOTEARS | 0.63±0.01 | 0.60±0.00 | 1.00±0.00 | 1.00±0.00 | 3.46±0.13 | 3.02±0.01 |
| $n_2=100$ | PC | 0.73±0.02 | 0.79±0.00 | 0.50±0.00 | 0.50±0.00 | 3.62±0.15 | 3.88±0.03 |
| ER Model | LiNGAM | 0.62±0.01 | 0.60±0.00 | 0.98±0.02 | 1.00±0.00 | 3.18±0.09 | 3.00±0.00 |
| S3 | NSCSL-TE | 0.10±0.02 | 0.01±0.00 | 0.94±0.02 | 1.00±0.00 | 0.84±0.16 | 0.06±0.02 |
| $p=5$ | NSCSL-DE | 0.08±0.02 | 0.01±0.00 | 0.84±0.03 | 0.81±0.00 | 1.30±0.14 | 1.00±0.00 |
| $n_1=30$ | NOTEARS | 0.11±0.02 | 0.01±0.00 | 0.96±0.02 | 1.00±0.00 | 0.76±0.16 | 0.06±0.02 |
| $n_2=100$ | PC | 0.00±0.00 | 0.00±0.00 | 0.66±0.02 | 0.80±0.00 | 1.68±0.12 | 1.00±0.00 |
| ER Model | LiNGAM | 0.03±0.01 | 0.00±0.00 | 0.98±0.01 | 1.00±0.00 | 0.24±0.09 | 0.02±0.01 |

The proof and detailed conditions for Thm. 5.1 are provided in App. D, which align with those commonly imposed in causal structural learning [e.g., 33]. Our proof follows similar strategies but accounts for the extra penalty term from causal effects. Notice that the explicit forms of causal effects under LSEM are linear combinations of elements of $\boldsymbol{B}$. This implies our new regulation can similarly vanish away as $n$ goes to infinity.

## 6 Experiments

**Experiment design.** Scenarios are generated as follows. We consider the dimension of variables in the graph as $p=5$ in Scenarios 1 to 3 (S1 to S3), $p=20$ in Scenario 4 (S4), and $p=50$ in Scenario 5 (S5), to examine the scalability of NSCSL. For each scenario, the DAG that characterizes the causal relationship among variables $\boldsymbol{O}=(\boldsymbol{Z},Y)$ is generated from the Erdős-Reńyi (ER) model with an expected degree as 2 for S1 to S3 and degree as 5 for S4 to S5. We also generate the graph from the scale-free (SF) model for S5 with the degree of 5 to examine the robustness of the proposed method under diverse synthetic graphs. Each edge is assigned positive weights. We set the last variable as the outcome of interest $Y$, and generate the data based on LSEM by $\boldsymbol{Z}=\boldsymbol{B}^\top\boldsymbol{Z}+\boldsymbol{\epsilon}$, where $\boldsymbol{\epsilon}$ is a random vector of jointly independent binary variables with equal probability taking value one or zero. Thus, the outcome is nonnegative and discrete. In addition, we also include a nonlinear structural equation model for S4 where $O_i := \psi_i\{\mathrm{PA}_{O_i}(\mathcal{G})\}+e_{O_i}$ and $\psi_i(x)=\lfloor 2\log(x+1)\rceil$ where $\lfloor x\rceil$ rounds to nearest integer for $x$. In S1, the true causal graph contains one spurious node (indexed by 0) and three non-spurious nodes (indexed by 1, 2, and 3), as shown in sub-figures (a) of Fig. E.7, with the associated true NSCG in sub-figures (b) of Fig. E.7. Moreover, we design a balanced setting with half spurious variables and half non-spurious variables in S2, as depicted in Fig. E.8, and a baseline setting without any spurious variables in S3, shown in Fig. E.9. Finally, S4 contains 2 non-spurious variables with 17 spurious variables in Fig. E.10. The experiments are conducted on a Google Cloud Platform virtual machine with 8 processor cores and 32GB memory.

**Methods and benchmark specification.** We apply the proposed NSCSL based on TE and DE as the criteria of necessity and sufficiency, respectively, to capture the marginal and conditional causal effect of the confounders. Note that we consider fully identifiable models in the experiments so that it is meaningful to evaluate causal effects from the estimated graph. The underlying causal structure learning algorithm is set to NOTEARS [46]. We also compare the proposed method against other methods, including PC [35] and LiNGAM [34] for S1 to S5; DAG-GNN [44], GES with generalized score [GSGES, 11], fast causal inference [FCI, 36], and the causal additive model [CAM, 4], for all high-dimensional settings in S4 and S5. Here, we use a graph threshold of 0.3 (commonly used in the literature [46; 44; 48; 5]) to prune the noise edges for a fair comparison. The training details are provided in Table E.1. The true and estimated graphs with the associated matrix under different approaches are illustrated in Figs. E.2 to E.6 and Figs. E.7 to E.11 in App. E for S1 to S4. The comparison results across different sample sizes ($n$) are presented in Table 2 for S1 to S3, in Table 3 for S4 to S5 with linear ER model, in Table 4 for S4 with nonlinear ER model, and in Table 5 for S5 with linear SF model. All the results are evaluated by false discovery rate (FDR), true positive

Table 3: Comparison studies under S4 to S5 under different sample sizes ($n$) and dimensions ($p$) with the Erdős-Reńyi (ER) model. Methods are evaluated by FDR, TPR, SHD, and runtime (seconds), with standard errors (SE) reported for each metric, over 50 replications.

| Scenario | Method | FDR±SE $n_1$ (small) | $n_2$ (large) | TPR±SE $n_1$ (small) | $n_2$ (large) | SHD±SE $n_1$ (small) | $n_2$ (large) | Time±SE $n_1$ (small) | $n_2$ (large) |
|---|---|---|---|---|---|---|---|---|---|
| S4 | NSCSL-TE | 0.11±0.02 | 0.00±0.01 | 1.00±0.00 | 1.00±0.00 | 0.86±0.24 | 0.04±0.01 | 10.8±0.3 | 56.2±1.1 |
| $p = 20$ | NSCSL-DE | 0.11±0.02 | 0.00±0.01 | 0.69±0.01 | 0.67±0.00 | 1.34±0.08 | 1.00±0.01 | 12.4±0.8 | 54.5±1.3 |
| $n_1 = 100$ | NOTEARS | 0.93±0.00 | 0.92±0.00 | 1.00±0.00 | 1.00±0.00 | 40.80±0.20 | 36.40±0.04 | 22.9±6.4 | 69.8±8.7 |
| $n_2 = 1000$ | PC | 0.92±0.00 | 0.95±0.00 | 0.51±0.02 | 0.67±0.00 | 19.08±0.17 | 33.34±0.03 | 6.9±0.5 | 16.3±0.8 |
| ER Model | LiNGAM | 0.92±0.00 | 0.93±0.00 | 0.99±0.01 | 1.00±0.00 | 33.00±0.20 | 37.10±0.02 | 8.1±0.5 | 24.6±0.6 |
| Degree=5 | DAGGNN | 0.93±0.00 | 0.93±0.00 | 0.97±0.01 | 0.97±0.00 | 41.34±0.19 | 40.10±0.06 | $28.3^2$±4.3 | $39.1^2$±7.2 |
| | GSGES | 0.98±0.01 | 0.98±0.00 | 0.22±0.02 | 0.20±0.01 | 43.20±0.26 | 45.70±0.34 | $14.9^2$±7.5 | $26.3^2$±9.1 |
| | FCI | 0.98±0.01 | 0.99±0.00 | 0.10±0.02 | 0.06±0.00 | 22.70±0.17 | 32.90±0.02 | 6.6±0.3 | 12.7±0.2 |
| | CAM | 0.93±0.00 | 0.94±0.00 | 0.63±0.02 | 1.00±0.00 | 29.80±0.38 | 40.30±0.07 | $19.6^2$±18.3 | $25.6^2$±23.2 |
| S5 | NSCSL-TE | 0.03±0.01 | 0.02±0.01 | 0.86±0.03 | 0.93±0.01 | 2.18±0.13 | 1.58±0.07 | 110.1±3.9 | $21.1^2$±12.1 |
| $p = 50$ | NSCSL-DE | 0.02±0.02 | 0.01±0.01 | 0.29±0.02 | 0.28±0.01 | 10.08±0.21 | 9.76±0.13 | 119.0±5.5 | $23.5^2$±10.9 |
| $n_1 = 1000$ | NOTEARS | 0.86±0.04 | 0.85±0.01 | 0.93±0.03 | 0.92±0.01 | 79.20±1.40 | 77.12±0.53 | 128.3±8.2 | $26.9^2$±15.1 |
| $n_2 = 3000$ | PC | 0.96±0.03 | 0.97±0.02 | 0.07±0.01 | 0.06±0.01 | 82.12±1.21 | 88.28±1.63 | 20.9±0.5 | 35.9±3.2 |
| ER Model | LiNGAM | 0.86±0.02 | 0.86±0.01 | 0.97±0.02 | 0.99±0.01 | 86.12±1.20 | 85.70±0.84 | 43.1±6.3 | 145.3±7.9 |
| Degree=5 | DAGGNN | 0.87±0.02 | 0.88±0.01 | 0.93±0.02 | 0.94±0.01 | 87.50±1.10 | 85.62±0.96 | $49.3^2$±7.5 | $81.1^2$±87.6 |
| | GSGES | 0.89±0.03 | 0.93±0.01 | 0.19±0.03 | 0.12±0.01 | 93.54±1.46 | 95.70±0.79 | $31.2^2$±10.1 | $45.3^2$±18.1 |
| | FCI | 0.96±0.02 | 0.97±0.01 | 0.08±0.01 | 0.07±0.01 | 84.00±0.80 | 88.50±0.60 | 14.3±0.8 | 17.7±0.5 |
| | CAM | 0.93±0.04 | 0.95±0.02 | 0.66±0.03 | 0.67±0.02 | 126.00±3.64 | 127.80±2.06 | $28.4^2$±31.7 | $75.6^2$±63.9 |

rate (TPR), and the structural Hamming distance (SHD) to the true causal graph, with their standard errors, over 50 replications. The average running time of these methods is also reported in Table 2 to Table 5 to reflect the computational complexities. In addition, the sensitivity analyses concerning all hyperparameters in Table E.1 are conducted using S4, as presented in Fig. 2.

**Results and conclusion.** NSCSL performs the best in discovering NSCG in S1, S2, S4, and S5, and shows comparably best results in S3 (a setting without any spurious variables). To be specific, the benchmark methods for causal structural learning aim to reveal causal relationships in the whole graph (i.e., sub-figures (a) in Figs. E.2 to E.6), which contains spurious effects on the target outcome $Y$. The proposed algorithm overcomes this drawback by purely identifying the true important causal relationships (i.e., sub-figures (b) in Figs. E.2 to E.6). By comparing the sub-figures (c) and (d) in Figs. E.2 to E.6 as well as Table 2 to Table 5, NSCSL-TE detects all necessary and sufficient causal paths towards the outcome, while NSCSL-DE only extracts direct causal relationships between the features and the outcome, resulting in a slightly lower TPR and slightly higher SHD than NSCSL-TE. Furthermore, from Table 2 to Table 5, the results under the proposed algorithm approach the truth more closely as the sample size increases in all scenarios, regardless of the underlying graph models and data-generating process. In contrast, the benchmark methods all fail to discover NSCG in the high-dimensional setting, exhibiting extremely high FDRs and SHDs. In addition, Table 3 to Table 5 reveal that NSCSL is as fast as the quickest benchmarks such as PC, LiNGAM, and FCI, and significantly faster than others like DAGGNN, GSGES, and CAM. Our method's integration of treatment effects into the optimization adds efficiency and restricts the searching space, making it practical and even beating NOTEARS in computation. The sensitivity analyses in Fig. 2 indicate that our method remains robust to these parameters set within a reasonable range.

**Real data analysis - Sachs et al. [30].** We conduct real data analysis using the benchmark data from Sachs et al. [30]. To validate our method's capacity to find the NSCG and align with Def. 3.2, we designated the protein Akt as the target outcome. This designation ensures that NSCG exists (see Fig. 3) and that finding an NSCG is meaningful. Our method (NSCSL-TE) and seven baseline methods were applied and evaluated against the true NSCG associated with the protein Akt. Table 6 shows that our method achieves the best performance in finding the NSCG concerning the protein Akt.

**Real data analysis - Brem & Kruglyak [2].** Furthermore, we apply NSCSL to gene expression traits in yeast [2] using a dataset of 104 yeast segregants with diverse genotypes. The goal is to identify genes, known as quantitative trait loci (QTLs), influencing the expression level (nonnegative) of the genetic variant YER124C, a daughter cell-specific protein involved in cell wall metabolism. Following a similar approach as in Chakrabortty et al. [6], we include 492 QTLs by filtering out genes with missing or low variability in expression levels. The total sample size is 262. Given the high-dimensional QTLs, constructing an NSCG with only causally relevant variables for the outcome of interest is essential. We apply NSCSL-TE and compare it with all baseline methods. The summarized results in Table 7 highlight our method's ability to identify relevant genetic influencers

Table 4: Comparison studies under the nonlinear structural equation model for S4 over 50 replications.

| Scenario | Method | FDR±SE | TPR±SE | SHD±SE | Time±SE |
|---|---|---|---|---|---|
| S4 | NSCSL-TE | 0.03±0.01 | 0.83±0.01 | 0.60±0.02 | 55.8±0.3 |
| $p=20$ | NSCSL-DE | 0.03±0.01 | 0.50±0.01 | 1.60±0.02 | 56.8±0.2 |
| $n=1000$ | NOTEARS | 0.91±0.01 | 0.83±0.01 | 35.90±0.04 | 56.5±0.7 |
| ER Model | PC | 0.99±0.01 | 0.12±0.01 | 44.12±0.03 | 15.7±0.2 |
| Degree=5 | LiNGAM | 0.93±0.01 | 0.70±0.00 | 37.30±0.03 | 11.6±0.1 |
| | DAGGNN | 0.94±0.01 | 0.86±0.01 | 34.80±0.06 | $41.3^2$±9.8 |
| | GSGES | 0.98±0.01 | 0.23±0.01 | 52.40±0.38 | $22.1^2$±7.5 |
| | FCI | 0.97±0.01 | 0.13±0.01 | 33.80±0.06 | 12.6±0.3 |
| | CAM | 0.95±0.01 | 1.00±0.01 | 31.60±0.08 | $27.9^2$±33.7 |

Table 5: Comparison studies under the scale-free (SF) model for S5 over 50 replications.

| Scenario | Method | FDR±SE | TPR±SE | SHD±SE | Time±SE |
|---|---|---|---|---|---|
| S5 | NSCSL-TE | 0.02±0.02 | 0.78±0.03 | 5.08±0.11 | 135.7±5.6 |
| $p=50$ | NSCSL-DE | 0.02±0.02 | 0.51±0.02 | 17.20±0.35 | 147.0±6.3 |
| $n=1000$ | NOTEARS | 0.88±0.04 | 0.75±0.03 | 123.10±1.50 | 160.5±8.1 |
| SF Model | PC | 0.97±0.03 | 0.06±0.02 | 79.34±1.17 | 32.1±0.8 |
| Degree=5 | LiNGAM | 0.91±0.02 | 0.98±0.01 | 212.00±5.12 | 117.7±6.3 |
| | DAGGNN | 0.92±0.02 | 0.85±0.02 | 203.50±7.80 | $57.3^2$±13.6 |
| | GSGES | 0.96±0.03 | 0.10±0.03 | 98.34±2.15 | $35.9^2$±13.5 |
| | FCI | 0.97±0.02 | 0.07±0.01 | 81.70±1.40 | 15.7±1.1 |
| | CAM | 0.98±0.04 | 0.24±0.03 | 218.00±8.15 | $37.1^2$±53.6 |

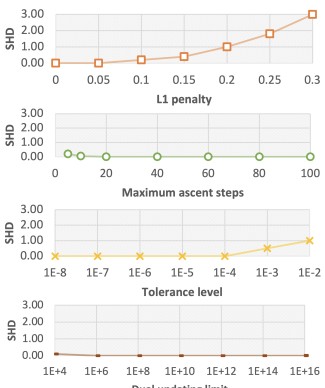

Figure 2: Sensitivity analyses.

Table 6: Real data results for the single-cell data by Sachs et al. [30], evaluated by total edges, correct edges, and SHD, based on the true NSCG with respect to the protein Akt.

| Method | NSCSL | NOTEARS | PC | LiNGAM | DAGGNN | GSGES | FCI | CAM |
|---|---|---|---|---|---|---|---|---|
| Total Edges | 8 | 20 | 25 | 6 | 33 | 28 | 24 | 7 |
| Correct Edges | 4 | 2 | 2 | 1 | 4 | 2 | 2 | 1 |
| SHD | 8 | 21 | 28 | 11 | 30 | 32 | 27 | 10 |

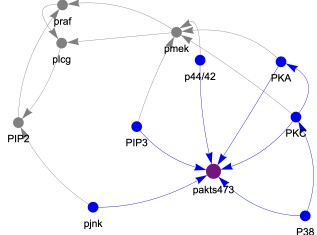

Figure 3: The causal signaling network in Sachs et al. [30], where the blue-colored sub-graph is the true NSCG for protein Akt (purple).

Table 7: Real data results for the yeast gene data, evaluated by total edges, the identified parents/ancestors of the variant YER124C, and the edges towards YER124C.

| Method | NSCSL | NOTEARS | PC | LiNGAM | DAGGNN | GSGES | FCI | CAM |
|---|---|---|---|---|---|---|---|---|
| Total Edges | 11 | 25 | 22 | 15 | 35 | 27 | 22 | 33 |
| # Parents/Ancestors of YER124C | 8 | 8 | 6 | 4 | 8 | 7 | 5 | 7 |
| # Edges towards YER124C | 11 | 9 | 8 | 6 | 10 | 9 | 8 | 10 |

for the variant YER124C without contamination by irrelevant genes. The estimated causal graph and causal effects as well as more detailed analyses are provided in App. E.2. These observations align with findings from our simulation studies, further supporting NSCSL's superiority in revealing important causal features.

## 7 Limitations and Future Research

In this work, we introduced NSCSL that leverages causal effects/POCs to systematically assess feature importance while learning a causal graph. By identifying a subgraph closely related to the outcome, our method filters irrelevant variables, presenting a significant advancement in the field. Extensive empirical evaluations on simulated and real-world data underscore NSCSL's superior performance over existing algorithms, including important findings on yeast genes and the protein signaling network. However, this promising advancement is not without limitations. First, NSCSL, like most existing causal structural learning methods, assumes no unmeasured confounders (A2) and the causal Markov condition. These assumptions may not hold in practice, leading to biased causal effect estimates and potential errors in the causal graph. Second, NSCSL employs absolute causal effects as a substitute for POCs to facilitate estimation in high-dimensional settings. Although theoretically consistent under certain conditions, examining the differences between these two methods in general feature and outcome spaces is an area for future research.

## Acknowledgments and Disclosure of Funding

Research reported in this publication was supported in part by the Amazon Web Services (AWS) Cloud Research program, the Office of Naval Research under grant N00014-23-1-2590, and the National Science Foundation under grant No. 2231174 and No. 2310831. The authors thank the area chair and the reviewers for their constructive comments which have led to a significant improvement of the earlier version of this article.

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
