# OpenReview forum: "On Learning Necessary and Sufficient Causal Graphs"
_NeurIPS.cc/2023/Conference — NeurIPS 2023 spotlight_

### Official Review · Reviewer_hThF · 2023-07-05

**Soundness:** 3 good
**Presentation:** 3 good
**Contribution:** 3 good
**Rating:** 6
**Confidence:** 3

**Summary:**

The paper introduces a novel method to identify causally relevant variables with respect to a specific target node. Leveraging the concept of probabilities of causation, the authors propose an approach that efficiently and systematically identifies a subgraph containing the relevant ancestors of the target node. This method has been evaluated using both artificial and real-world datasets.

**Strengths:**

* The paper is well written and motivated. In particular, the problem setting is nicely introduced.
* Comprehensive mathematical definitions and explanations.
* The method is relevant for different causal inference problems.

See the Question section for further remarks.

**Weaknesses:**

* The paper has a slightly confusing mixture of the potential outcome (PO) framework (for instance, using A1 and A2) and the graphical causal model framework. The notation from the PO framework doesn't seem essential here, especially considering that the majority of the work is based on graphical causal models.
* It seems there is an implicit assumption that the target variable of interest is a leaf node in the graph (i.e., it has no descendants). While this aligns with the typical PO causal effect estimation setup, the proposed method could also be beneficial for other causal inference questions beyond effect estimation.
* Unclear if the proposed method is scale invariant, which would be an important property to avoid arbitrary changes in the result due to (often) unknown rescalings. This concern mostly stems from the references to NOTEARS, which is not scale invariant (see “Unsuitability of NOTEARS for Causal Graph Discovery” by Kaiser et al.). This could be discussed more clearly.
* Although the empirical results are convincing, the number of baseline methods is quite limited. Here, more and newer causal discovery algorithms could be included in the evaluation.

See the Question section for additional points and remarks.

**Questions:**

A few general remarks and questions:


* The abstract is excellent, being very concise, nicely motivating the problem, and describing the goal of the paper.
* The title could be more precise in mentioning that it pertains to a specific target variable, especially since causal discovery algorithms typically don’t have that (simplifying) restriction.
* The beer and diapers example is engaging, but it implies that we either do not observe the hidden confounder or would incorrectly learn a model that includes "diapers" to predict "beer". Is this meant as an example of including unnecessarily many variables or as an example where causal discovery fails due to hidden confounders?
* Related to the previous point, it appears that the authors have a typical PO setting of "treatment, covariates, and outcome" in mind. I might have missed this detail, but for instance, it seems (implicitly) assumed that no variable is a causal child of the target. The general setting could be introduced more clearly.
* The scoring method appears to target effect estimation tasks. If that is indeed the causal inference question you're aiming for, this should be discussed more explicitly. Generally, the proposed methods might also be interesting for other causal questions, such as contribution or root cause analysis.
* The works “Quantifying causal influences” and “Quantifying intrinsic causal contributions via structure-preserving interventions” by Janzing et al. might provide an interesting alternative for more general (and non-linear) measures for quantifying causal influences.
* The limitation to a discrete target variable is relatively stringent. Perhaps the previously mentioned work could inspire ideas on how to generalize this further in follow-up research.
* A minor point: The edges D_X are not formally introduced in detail.
* In Definition 3.1, does this only include direct parents and grandparents but not all other (potentially further) ancestors?
* In the discussion of Example 4.7, it's somewhat confusing why one would include X_D in the first place if the graph is known. And if it is unknown, why would one even "blindly" include all variables in the model, considering that there could also be variables that are causal children of the target? Again, this seems to implicitly assume that the target node of interest is a leaf node.
* Section 5 is insightful, effectively blending the mathematical details with the overall procedure.
* It's unclear if your method only works for linear relationships/linear structural causal models (SCMs). Could you provide clarification?
* Overall, the experimental results are compelling, but the selection of baseline methods could be broader given the significant advances in recent causal discovery approaches.


--Update after rebuttal--
I have read the rebuttal and further discussed with the authors. Since there are several smaller points that need to be revised in the final version, I stick to my initial (positive) score.

**Limitations:**

There are no concerns regarding societal impact. For technical limitations, refer to the points raised in the Questions and Weakness sections.

---

> ### Author Rebuttal · Authors · 2023-08-10
>
> Thank you for your valuable comments and suggestions! We are honored by the reviewers’ recognition of our well-motivated and nicely introduced setting, comprehensive mathematical definitions/explanations, and utilities of our method. We have diligently addressed all your questions and comments. Below, we summarize your questions and comments in quotes, followed by our point-by-point responses. Please refer to **the one-page PDF in the general response (GR)** for all additional simulation/real-data results.
>
> 1. > Mixture of the potential outcome (PO) and the graphical causal model
>
> **Re:** Thanks for this keen comment. We use the notations of the PO framework intentionally to interconnect POCs and causal effects. We first employed the PO framework to construct the POCs in Section 4.1 for a generalized multi-variable setting within a causal graph. Then we introduce causal effects in Section 4.2, grounding these concepts in the same PO framework, and establish the theoretical relationship between these two sets of concepts. In Section 5, we integrate the PO and graphical causal model frameworks by employing causal effects as a regulator to guide the causal discovery process of finding NSCGs.
>
> 2. > Assume target variable of interest is a leaf node?
>
> **Re:** We greatly value this incisive question and clarify that our framework **doesn’t necessitate that the target variable be a leaf node**. In Definitions 3.1-3.2, the NSCG for a target variable contains only its parents and ancestors, not its children (if any). Our algorithm is flexible to contain a causal identification constraint (lines 269-271) to utilize if the target variable is a leaf node or drop this constraint where such information is unknown. To support our arguments, we have run additional real data analysis using data from Sachs et al. (2005), where we designated the protein Akt as the target (see **Figure 2 in GR**) but **removed the causal identification constraint**. As shown in **Table 4 in GR**, our method achieves the best performance in finding the NSCG for Akt.
>
> 3. > Is scale invariant?
>
> **Re:** Thanks for this excellent question. NSCSL is scale-invariant when we appropriately choose the causal discovery base learner and model the treatment effects/POCs. Though NOTEARS lacks scale invariance, our method's flexibility allows for the integration of scale-invariant methods like PC/LinGAM. Additionally, under LSEM, the rescaling will not affect the relative rank of the features based on causal effects. In the nonlinear case, we proposed to use POCs which by their definitions are scale-invariant.
>
> 4. > More and newer causal discovery algorithms
>
> **Re:** Thanks for this excellent suggestion. We've included four new baselines, including DAG-GNN, GSGES, FCI, and CAM. The new comparisons encompass Scenario 4 ($p=20$, $n=100,1000$) and new Scenario 5 ($p=50$, $n=1000,3000$), under varied settings. As displayed in **Tables 1-3 in GR**, our method outperforms all baseline methods.
>
> 5. > Title precision
>
> **Re:** We greatly appreciate this suggestion and propose a refined title: *On Learning Necessary and Sufficient Causal Graphs for Target Variables*.
>
> 6. > The beer and diapers example clarification
>
> **Re:** Yes, this is an example of including unnecessarily many variables of "diapers" that are spuriously related to predicting "beer".
>
> 7. > Assume no variable is a causal child of the target?
>
> **Re:** Please refer to point #2.
>
> 8. > Scoring method and target effect estimation
>
> **Re:** We acknowledge that our method primarily aims at estimating causal effects or POCs for feature selection and added that our approach could extend to diverse causal inquiries.
>
> 9. > Works by Janzing et al. as an alternative
>
> **Re:** We've added Janzing et al. (2013 & 2020) that indeed shed light on causal feature selection in nonlinear contexts, to our related works. Unlike these works requiring a known graph, our method uniquely integrates causal graph learning with feature selection.
>
> 10. > Limitation to a discrete target variable
>
> **Re:** The necessity for a discrete target variable is due to the meaningfulness of conditional probabilities in Definitions 4.1-4.3. Yet, we recognize that POCs can be extended to continuous outcomes with proper density functions of necessity and sufficiency, though estimating such concepts can be more challenging in practice. Our current choice simplifies the presentation and computation.
>
> 11. > Definition of $D_X$
>
> **Re:** The edge set $D_X$ (see line 91) encompasses all the directed edges in the causal graph for the node set $X$. We’ve added more details.
>
> 12. > Definition 3.1 scope
>
> **Re:** It not only includes direct parents and grandparents but all other (potentially further) ancestors, based on $PA_Y(\mathcal{G})$ defined in line 95.
>
> 13. > Why include $X_D$ in Example 4.7
>
> **Re:** Our work learns NSCG in an unknown causal graph, so the inclusion of $X_D$ is indeed possible. Second, $X_D$ may be incorporated due to its strong correlation with $X_B$ in marketing reports. The practice of including many confounders aligns with the goal to learn the sufficient causal graph (see lines 22-25). Concerning the target node being a leaf node, please refer to point #2.
>
> 14. > Applicability to linear/nonlinear SCMs
>
> Re: Our method indeed applies to both linear and nonlinear SCMs. Section 5 outlines the main algorithm for linear SCMs, while an iterative algorithm for nonlinear cases is provided in Appendix A.2. We have tested additional non-linear SCMs as reported in **new Table 2 of GR**, which shows our method consistently outperforms all baselines.
>
> We would like to sincerely thank the reviewer for carefully reviewing our paper and recognizing our efforts! We have striven to address all concerns comprehensively, with all clarifications and discussions integrated into the revised manuscript. We would be happy to address further comments or suggestions if there are any and we look forward to hearing from you soon.

---

> > ### Comment · Reviewer_hThF · 2023-08-11
> >
> > I would like to thank the authors for their thoughtful response and detailed answers to the questions I raised.
> >
> > The results from the new set of experiments are truly impressive. I might be misunderstanding something in the tables, but it seems that the other algorithms, based on their high FRD and SHD, performed significantly worse to the extent that they appear almost useless. Could you briefly comment on this and confirm there isn’t an error?

---

> > > ### Author Response · Authors · 2023-08-12
> > > **Response to Official Comment by Reviewer hThF**
> > >
> > > We sincerely appreciate your timely feedback and recognition of our response as well as the newly-conducted experiments. We understand your concerns about the observed high FRD and SHD in the tables, and we would like to clarify our experiment results and the evaluation metrics as follows.
> > >
> > > 1. **Evaluation Based on NSCG**: Our experiment results are evaluated against the true necessary and sufficient causal graphs (NSCGs) derived from Definition 3.2, a sub-structure of the full graph, not the full graphs themselves. The details, including the illustrations of true NSCGs and full graphs, can be found in Sections D.2 and D.3 in Appendix, where the true NSCG contains much fewer edges than the full graph.
> > >
> > > 2. **Baseline Performance**: We acknowledge that the selected baseline methods are capable of identifying the full causal graph given their model assumptions are met, however, they are less effective in finding the NSCGs, which is our primary focus in this study.
> > >
> > > 3. **High FDR and SHD Explanation**: The observation of high FDR and SHD stems from the baselines identifying many spurious/irrelevant nodes/edges that aren't part of the NSCG for the target outcome of interest. For example, in Scenario 4, the discrepancy between the true NSCG (with only 3 edges) and the full graph (containing over 35 edges) leads to an unavoidable SHD higher than 30.
> > >
> > > 4. **Proposed Method Performance**: Our method excels in recovering the true NSCG, not the full causal graph. Thus, while it demonstrates superior performance in this context, it would not outperform the baselines if the goal were to recover the full graph. We've clarified the evaluation metrics in our revised manuscript.
> > >
> > > We hope these explanations adequately address your concern. We remain committed to enhancing our paper and are open to further questions or comments. Once again, thank you for your invaluable contribution to our work!

---

> > > > ### Comment · Reviewer_hThF · 2023-08-16
> > > >
> > > > Thank you for the explanation. While I don't have a concrete suggestion, I am wondering if there is an experimental setup that could be more fair in the baseline comparison. Since there are several smaller points that need to be revised in the final version, I will stick to my initial score and believe that this is a good paper.

---

> > > > > ### Author Response · Authors · 2023-08-17
> > > > > **Response to Official Comment by Reviewer hThF**
> > > > >
> > > > > We are very grateful for your continued engagement and kind acknowledgment of our work as a good paper. Your thoughtful comments on the baseline comparison have prompted us to make the following clarifications:
> > > > >
> > > > > 1. **A Fair Comparison - Scenario 3**: In our original paper, Scenario 3 (S3) serves as a baseline setting where there are no spurious variables in the causal graph (as illustrated in Figure D.8 in the Appendix). In this scenario, learning the necessary and sufficient causal graph is equivalent to learning the full causal graph. Based on Table 2 in our original paper, the proposed NSCGL-TE performs comparably to the best benchmarks but does not surpass them. Moreover, NSCGL-DE has a constant bias because it prunes indirect edges toward the outcome of interest. We included this scenario to demonstrate the worst-case performance of our NSCG learning method, ensuring a fair comparison.
> > > > >
> > > > > 2. **Additional Metrics for Evaluation**: As elaborated in our updated real data analysis, we have provided additional evaluation metrics as shown in **new Tables 4-5 in our general response**. These include the correct edges and edges towards the outcome of interest, reflecting the method's capacity to learn the sufficient causal graph. In other words, if a causal discovery method correctly identifies the full causal graph, it should also identify all edges within the NSCG. We believe these supplementary metrics will contribute to a more balanced and fair evaluation, aligning with your insightful recommendation.
> > > > >
> > > > > We sincerely hope that these clarifications have addressed your comments to your satisfaction. We remain devoted to refining our paper and are open to any further questions or insights you may have. We thank you once again for your vital contributions to our work!

---

### Official Review · Reviewer_Qe9n · 2023-07-06

**Soundness:** 3 good
**Presentation:** 2 fair
**Contribution:** 3 good
**Rating:** 6
**Confidence:** 4

**Summary:**

The paper deals with proposing a necessary and sufficient causal graph that explicitly only model the causal variables required rather than the complete causal graphs that can be inefficient and also introduce spurious correlations between variables.

**Strengths:**

1. Important problem, well-defined and well written. The NSC is intuitive and makes perfect sense.

2. Using or rather extending the POC concept to assess the necessity and sufficiency of features in determining the outcome is very interesting.



**Weaknesses:**

1. I have some concerns regarding the scalability of the method and the max number of samples considered is 100. Also it would have been ideal if some real world example was shown and experimented with.

2. Assuming Markovian condition limits the overall applicability of the methods especially in the real world scenarios.

3. Conclusion section reduced to only limitations and future work hindering the overall completeness of the paper. I know this would have been done due to the space limit but in my personal opinion every paper should have a proper conclusion section.

**Questions:**

1. Can the authors comment on how the Markovian assumption limits the applicability of their method?

2. Another factor is that the variables considered are only binary. This again effects the applicability and it would be nice  to see some discussion on this.

3. Considering 100 samples does not explicitly show the scalability of the method. Can some more large scale experiments be shown?

Overall a nice paper with a simple yet nice idea but not without its flaws. I lean towards acceptance as of now but have concerns about the applicability of the method in real world scenarios and will wait for the authors replies to make a final decision.

**Limitations:**

No concerns here.

---

> ### Author Rebuttal · Authors · 2023-08-09
>
> Thank you for your valuable comments and suggestions! We greatly appreciate the reviewers’ acknowledgment that our work is an “important problem, well-defined and well written”, our proposed NSCG “is intuitive and makes perfect sense”, and that “using or rather extending the POC concept to assess the necessity and sufficiencyis very interesting”. We have carefully addressed all your questions and comments. In the following, your questions and comments are summarized in quotes, followed by our point-by-point responses. Please refer to **the one-page PDF in the general response (GR)** for all additional simulation/real-data results.
>
> 1. > …scalability of the method and the max number of samples considered is 100... Can some more large scale experiments be shown?
>
> **Re**: Thank you for this constructive comment. We have added further simulation results, with the number of nodes increased to 50 (as new Scenario 5) and the sample size increased to 1000 and 3000 for Scenarios 4-5. As shown in **Tables 1-3 in GR**, our method excels in these enhanced settings which highlights our method's practical applicability.
>
> 2. > ...ideal if some real world example was shown and experimented with.
>
> **Re**: We value this comment and have conducted additional real data analysis using the benchmark data from Sachs et al. (2005). To validate our method's capacity to find the NSCG and align with Definition 3.2, we designated the protein Akt as the target outcome. This designation ensures that NSCG exists (see **Figure 2 in GR**) and that finding an NSCG is meaningful. Our method and seven baseline methods were applied and evaluated against the true NSCG associated with the protein Akt. **Table 4 in GR** shows that our method achieves the best performance in finding the NSCG concerning the protein Akt.
>
> 3. > Assuming Markovian condition limits the overall applicability of the methods.
>
> **Re**: Thank you for this valuable comment. We agree that the causal Markovian condition can be violated in some real-world applications such as unmeasured confoundings, and also would like to emphasize that this is also a limitation of most causal discovery algorithms. Without the causal Markovian condition, we can only get up to a partial graph. To provide a precise but sufficient explanation in causal graphical contexts to an outcome of interest, which is the main goal and motivation of our paper, we require a causal Markovian condition. However, we acknowledge the possible practical limitations of required such an assumption, and the extensions are of particular interest. We have included this in our open discussion as well.
>
> 4. > Conclusion reduced to only limitations and future work…every paper should have a proper conclusion.
>
> **Re**: We are very grateful for this excellent suggestion. The revised conclusion of our paper is as follows:
>
> *In this work, we introduced NSCSL that leverages causal effects/POCs to systematically assess feature importance while learning a causal graph. By identifying a subgraph closely related to the outcome, our method filters irrelevant variables, presenting a significant advancement in the field. Extensive empirical evaluations on simulated and real-world data underscore NSCSL's superior performance over existing algorithms, including important findings on yeast genes and the protein signaling network.*
>
> *However, this promising advancement is not without limitations. First, NSCSL, like most existing causal structural learning methods, assumes no unmeasured confounders (A2) and the causal Markov condition. These assumptions may not hold in practice, leading to biased causal effect estimates and potential errors in the causal graph. Second, NSCSL employs absolute causal effects as a substitute for POCs to facilitate estimation in high-dimensional settings. Although theoretically consistent under certain conditions, examining the differences between these two methods in general feature and outcome spaces is an area for future research.*
>
> 5. > …variables considered are only binary…nice to see some discussion on this.
>
> **Re**: Thank you for your insightful comments. Allow us to clarify that in our approach, the features $Z$ can encompass either discrete or continuous variables, while the outcome $Y$ is permitted to be a discrete random variable, not exclusively binary. Please refer to lines 99-101 for detailed information on this data structure flexibility.
>
> Our proposed POCs indeed generalize the bi-variable and binary setting found in Tian & Pearl (2000) (see lines 156-157). The stipulation for a discrete outcome variable ensures that the conditional probabilities in Definitions 4.1-4.3 are meaningful. In our simulations, we selected the noise variables to be binary, thus creating discrete data for both features and outcomes (lines 297-299). Moreover, the binary requirement, mentioned solely for establishing theoretical consistency between causal impact based on treatment effects and evaluation by POCs (Theorem 4.6, lines 197-198), does not constrain our NSCFL in practice.
>
> Finally, we acknowledge that the proposed POCs could be extended to handle continuous outcomes by defining an appropriate density function of necessity and sufficiency. However, estimating such a concept would be more challenging in practice relative to our current approach, which relies on discrete outcomes. We have chosen this path as it simplifies the presentation and facilitates a more practical analysis of causal relationships.
>
> We would like to sincerely thank the reviewer for reviewing our paper! We have strived to address all the reviewers' concerns appropriately. All the above clarifications and discussions have been incorporated into the revised paper. We would be delighted to entertain further comments or suggestions if any, and we eagerly anticipate your feedback.

---

> > ### Comment · Reviewer_Qe9n · 2023-08-12
> > **Thank you for your response**
> >
> > I would like to thank the authors for their response and the new results. My concerns are clarifies and thus I have raised my score accordingly to 6.

---

> > > ### Author Response · Authors · 2023-08-13
> > > **Thank you note to Reviewer Qe9n**
> > >
> > > Thank you very much for your kind acknowledgment of our response and the newly-added results. We are delighted to hear that your concerns have been clarified, and we appreciate the increased score. If there are any further questions or areas of interest, please don't hesitate to reach out. We remain committed to engaging with your insights and making our work as strong as possible. Thank you again for your thoughtful review and support!

---

### Official Review · Reviewer_LXJs · 2023-07-26

**Soundness:** 2 fair
**Presentation:** 3 good
**Contribution:** 2 fair
**Rating:** 6
**Confidence:** 2

**Summary:**

This paper is concerned with learning causal graphs from observational data. In particular, the authors propose to learn a subgraph of the full causal graph, which they refer to as necessary and sufficient causal graphs (NSCGs). They propose an algorithm which measures conditional probabilities of causation between variables to measure the causal effect of some treatment variable on the target variable. They evaluate their algorithm on synthetic and one real dataset and compare its performance against three other popular CD algorithms.

**Strengths:**

The paper is well written and interesting to read. The authors provide illustrative examples that help understand the theory and practical implications!

**Weaknesses:**

-	Not all causal discovery algorithms assume causal sufficiency, e.g. FCI [A] and related algorithms, output a CPDAG without assuming causal sufficiency. There exists some other notable work on this topic [e.g. B, C]
-	I find the experimental evaluation on synthetic data to be too limited. The authors compare their method only to three other Causal Discovery Algorithms (including NOTEARS, which is actually not suitable for causal discovery because of missing scale-invariance [D]). In particular, they do not compare their method to algorithms that do not assume causal sufficiency (e.g. FCI).
-	On the one real-world dataset, the authors compare their method to NOTEARS only.
-	I find it difficult to follow their arguments, why the proposed algorithm performs better than NOTEARS on the real-world dataset. Statements like “This gene is required for sulfur amino acid synthesis” (L. 349) need a reference and further explanation.


[A] P. Spirtes, C. Glymour, and R. Scheines, Causation, Prediction, and Search. The MIT Press, 2001. doi: 10.7551/mitpress/1754.001.0001.

[B] J. M. Ogarrio, P. Spirtes, and J. Ramsey, “A Hybrid Causal Search Algorithm for Latent Variable Models,” in Proceedings of the Eighth International Conference on Probabilistic Graphical Models, PMLR, Aug. 2016.

[C] R. Bhattacharya, T. Nagarajan, D. Malinsky, and I. Shpitser, “Differentiable Causal Discovery Under Unmeasured Confounding,” Proceedings of the International Conference on Artificial Intelligence and Statistics, vol. 130, Apr. 2021.

[D] M. Kaiser and M. Sipos, “Unsuitability of NOTEARS for Causal Graph Discovery when Dealing with Dimensional Quantities,” Neural Processing Letters, vol. 54, no. 3, 2022.


**Questions:**

-	I found a small typo in L. 105: “while keeping the rest [of the] model unchanged”
-	Could the authors comment on whether they think using the Structural Intervention Distance (SID) [E] would be a meaningful additional metric for their empirical evaluation?
-	Why do the authors evaluate their algorithm on a single real-world dataset only? Why did they choose this particular dataset and not one of the more famous ones, e.g. [F]?
-	Can the authors comment on whether their algorithm is scale-invariant, given NSCSL uses absolute causal effects?


[E] Jonas Peters and Peter Bühlmann. Structural intervention distance for evaluating causal graphs. Neural Computation, 27(3):771–799, 2015.

[F] K. Sachs, O. Perez, D. Pe’er, D. A. Lauffenburger, and G. P. Nolan, “Causal Protein-Signaling Networks Derived from Multiparameter Single-Cell Data,” Science, vol. 308, no. 5721, pp. 523–529, Apr. 2005.


**Limitations:**

The authors provide an open discussion on the limitations of their proposed algorithm, which is greatly appreciated! Another limitation might be the strong assumptions on conditionals, however, this is a limitation of most causal discovery algorithms.

---

> ### Author Rebuttal · Authors · 2023-08-09
>
> Thank you for your valuable comments and suggestions! We're gratified by the reviewers’ recognition of our paper as well-written and interesting, with illustrative examples and an open discussion on limitations. We have carefully addressed all your questions and comments. In the following, your questions and comments are summarized in quotes, followed by our point-by-point responses. Please refer to **the one-page PDF in the general response (GR)** for all additional simulation/real-data results.
>
> 1. > Not all causal discovery algorithms assume causal sufficiency, e.g. FCI [A]…[B, C]
>
> **Re:** Thanks for your insightful comment. We agree that not all causal discovery algorithms assume causal sufficiency, which only yields a partial graph. Yet, we require causal sufficiency (i.e., A2) to find a precise and adequate causal graph to explain the target outcome. We recognize this potential limitation and the extensions to works such as [A, B, C] are intriguing. Specifically, we have outlined an iterative algorithm in Appendix A.2, which employs an arbitrary causal discovery method and subsequently estimates POCs until convergence, thereby allowing the integration of FCI without assuming causal sufficiency.
>
> 2. > ...experimental evaluation...too limited
>
> **Re:** This very constructive comment led us to add more simulations, with the number of nodes increased to 50 (new Scenario 5) and the sample size increased to 1000 and 3000 for Scenarios 4-5. As shown in **Tables 1-3 in GR**, our method excels in these enhanced settings which highlights our method's practical applicability.
>
> 3. > ...compare...only to three…NOTEARS…missing scale-invariance [D]...not compare…FCI
>
> **Re:** Thanks for this excellent suggestion. We've expanded comparison studies with four additional baselines, including FCI, DAG-GNN, GSGES, and CAM. The new comparisons encompass S4 ($p=20$, $n=100,1000$) and new S5 ($p=50$, $n=1000,3000$), under varied settings. As shown in **Tables 1-3 in GR**, our method outperforms all baselines. Additionally, we acknowledge NOTEARS' missing scale invariance but emphasize that our method’s flexibility allows integration with various causal discovery methods such as FCI aforementioned.
>
> 4. > ...real...compare...to NOTEARS only
>
> **Re:** Your constructive comment led us to conduct additional real data analysis using all baseline methods. The summarized results in **Table 5 in GR** highlight our method's ability to identify relevant genetic influencers for the variant YER124C without contamination by irrelevant genes. See also the response to point #9 for the additional analysis of data from Sachs et al. (2005).
>
> 5. > ...why…better than NOTEARS…L349...need…explanation
>
> **Re:** We appreciate this insightful suggestion and have included more references and explanations. Here, YLR303W is essential for sulfur amino acid synthesis[1]. And YER124C of interest is a daughter cell-specific protein involved in cell wall metabolism[2]. It has been shown that sulfur amino acid synthesis can influent cell wall metabolism[5][6]. These indicate that NSCGL which additionally identified YLR303W performs better than NOTEARS.
>
> [1] Brzywczy J (1993) Role of O‐acetylhomoserine in sulfur amino acid synthesis. Yeast.
>
> [2] Colman-Lerner A (2001) Yeast daughter-specific genetic programs. Cell.
>
> [3] Takahashi H (2001) Sulfur economy and cell wall biosynthesis. Plant physiology.
>
> [4] de Melo (2019) The regulation of the sulfur amino acid. Scientific Reports.
>
> 6. > typo in L 105
>
> **Re:** Thank you so much for pointing this out. We’ve corrected this typo.
>
> 7. >...comment on.. the Structural Intervention Distance (SID) [E]
>
> **Re:** We truly appreciate this constructive suggestion. Indeed, we acknowledge that SID is a noteworthy metric for causal discovery evaluation. We believe that adding SID could significantly enhance our empirical studies. Unfortunately, the SID R package is currently unavailable in CRAN (see https://cran.r-project.org/web/packages/SID/index.html), and other tools like 'cdt' rely on this R package to compute SID. We are in the process of implementing SID into Python, but this effort is beyond the scope of the current rebuttal period. We are committed to including this metric once it becomes available.
>
> 8. > Why…evaluate…on a single real dataset only? Why…not…[F]?
>
> **Re:** Thank you for this insightful comment. We have run additional real data analysis using the benchmark data from Sachs et al. (2005). To align with Definition 3.2, we designated the protein Akt as the target outcome. Our method and all baselines were applied and evaluated against the true NSCG for Akt (see **Figure 2 in GR**). **Table 4 in GR** shows that our method achieves the best performance in finding the NSCG.
>
> 9. > ...is scale-invariant, given NSCSL uses absolute causal effects?
>
> **Re:** We appreciate this excellent inquiry. NSCSL is scale-invariant when we appropriately choose the causal discovery base learner and model the treatment effects/POCs. Though NOTEARS lacks scale invariance, our method's flexibility allows for the integration of scale-invariant causal discovery methods like NSCGL with FCI. Additionally, under LSEM, the rescaling will not affect the relative rank of the features based on absolute causal effects. In the nonlinear case, we proposed to use POCs which by their definitions are scale-invariant.
>
> 10. > ...assumptions on conditionals
>
> **Re:** We agree with your observation and appreciate your acknowledgment that assumptions on conditionals are indeed strong but commonly represented in most causal discovery algorithms. We have taken care to include this point in our open discussion.
>
> We would like to sincerely thank you for reviewing our paper! We have tried to address all your concerns in a proper way. All the above clarifications and discussions have been included in the revised paper. We would be happy to address further comments or suggestions if there are any and we look forward to hearing from you soon.

---

> > ### Author Response · Authors · 2023-08-15
> > **Eagerly Looking Forward to Feedback on Our Response**
> >
> > We sincerely appreciate the time and effort you've devoted to reviewing our work, and to providing many valuable and insightful feedback!
> >
> > Following your constructive suggestions, we have conducted five additional sets of experiments and have further clarified our assumptions,  base learner, scale-invariance, and real data evaluation. All of these details can be found in our response and within the one-page PDF file.
> >
> > We sincerely hope our further clarifications and experiments can fully address your concerns and can be helpful in the evaluation of our work. We are eagerly looking forward to your kind feedback!

---

> > > ### Comment · Reviewer_LXJs · 2023-08-16
> > >
> > > I would like to thank the authors for their detailed response. I appreciate the additional experiments, particularly on the Sachs et al. dataset. Since my concerns are properly addressed I'll increase my score.

---

> > > > ### Author Response · Authors · 2023-08-17
> > > > **Thank you note to Reviewer LXJs**
> > > >
> > > > We sincerely thank you for your continued engagement and kind acknowledgment of our detailed response, additional experiments, and real data analysis. We are thrilled that your concerns have been properly addressed, and we deeply appreciate the increased score. Once again, thank you for your constructive and encouraging feedback!

---

### Official Review · Reviewer_hnpE · 2023-07-26

**Soundness:** 3 good
**Presentation:** 3 good
**Contribution:** 2 fair
**Rating:** 6
**Confidence:** 3

**Summary:**

This paper studies the problem of feature selection when performing causal discovery and contributes to the limited literature in this field. Given a set of features, the main goal is to learn a causal graph from a subset of these features such that the learned graph only contains features that are “necessary and sufficient” for explaining an outcome of interest. This paper then develops two notions of quantifying the necessary and sufficient features using POC (probability of causation) and DE (direct effect)/TE (total effect). Given the measures to quantify the relevant features, they formulate a learning algorithm that jointly selects these relevant features (including the outcome of interest) and learns the causal graph. They also experimentally verify their method on four synthetic and one real-world dataset and show improvement over other baselines.


**Strengths:**

1. **Clarity**: This paper is clearly written and easy to follow. Though there is one limitation to the motivation of using POC (probability of causation) as the main metric to quantify the spurious features (see comment1 in the weakness part of the review).
2. **Significance**: This paper brings the notion of variable selection when learning the causal graph which is less explored in the literature.
3. **Theory:** Theorem 4.6 connects the two possible quantities i.e. POC and DE/TE, that can be used to quantify the spuriousness of each other and gives an expression for their lower bound in terms of marginals from the observational distribution which is novel.
4. **Experiment**: Across synthetic and real datasets one of their method i.e. NSCSL-TE shows consistent improvement compared to other baselines considered over different considered metrics (FDR, TPR, and SHD).


**Weaknesses:**

1. **Why POC instead of directly considering DE/TE**: Shouldn’t all the features with non-zero DE/TE on the outcome of interest should be considered in the final selected graph? Why POC is more fundamental than DE/TE is not properly motivated. I understand that both quantities are related (as stated in Theorem 4.6) but if the goal is to remove the spurious features when creating the causal graph then shouldn’t we directly use the TE/DE?
2. **Theory**: There is no guarantee that selecting the features based on the POC/DE/TE will converge towards the actual set of necessary and sufficient features as admissible by Definition 3.2.
3. **Experiment**: The synthetic setup considered in the paper assumes linear SEM as the data-generating process that might be limiting for the real-world DGPs (data-generating process). In the appendix, the author extends their algorithm for non-linear DGPs. It would be interesting to see some results on synthetic datasets with non-linear DGPs that could further attest to the applicability of their method in real-world scenarios in addition to one real-world dataset already considered in the paper.


**Questions:**

1. **Theory**: Line 125-126 state that Definition 3.1 refer to the sub-structure in the whole graph $G_{O}$ containing directed edges or path towards Y. It is not clear how the constraint on conditional probability distribution in Definition 3.1 allows $G_{V}$ to have features/nodes from $G_{O}$ that have a path towards Y. An example or an explanation will be helpful.
2. **Experiment**: Why does TPR decrease for NSCSL-DE for the S2-S4 on increasing the sample size?

---

> ### Author Rebuttal · Authors · 2023-08-09
>
> Thank you for your insightful comments and suggestions. We greatly appreciate your acknowledgment that our work "of variable selection when learning the causal graph is less explored in the literature", "Theorem 4.6 connects the two possible quantities is novel", our method "shows consistent improvement compared to other baselines", and our paper "is clearly written and easy to follow". We have carefully addressed all your questions and comments. In the following, your questions and comments are summarized in quotes, followed by our point-by-point responses. Please refer to **the one-page PDF in the general response (GR)** for all additional simulation/real-data results.
>
> 1. > Why POC instead of directly considering DE/TE?
>
> **Re:** Thank you for this valuable comment. POC, initially proposed by Pearl et al. (2000), assesses both necessity and sufficiency in a bi-variable, binary setting. In contrast, DE/TE quantifies the impact of treatment on the outcome by increasing it by one unit (see Pearl et al. (2009)). While we acknowledge that non-zero DE/TE should indeed be regarded as necessary, our contribution lies in connecting these two different concepts and demonstrating their equivalence under certain conditions. We emphasize that we are not arguing for the superiority of POC over DE/TE; rather, we find POC more conventional for introducing necessity and sufficiency in the existing literature (see Pearl et al. (2000), Tian & Pearl (2000), and Wang & Jordan (2021)).
>
> 2. > There is no guarantee that selecting the features based on the POC/DE/TE.
>
> **Re:** Thank you for this insightful comment. The challenge in developing a theory for the selected features is multi-faceted. Our method's unique aspect is that we learn the causal graph while selecting causal features, meaning the consistency of the selected features depends on the causal graph estimation's consistency. Further, developing a post-estimation selection is challenging, and the ambiguity in defining consistency arises from balancing losses of causal discovery and lower bounds of POCs/causal effects. While this presents a complex theoretical task, we provide further insights by establishing the consistency of estimated causal graphs:
>
> ***Theorem A***:  *Assume that $B$ and $O$ follow LSEM with independent Gaussian error and equal variance and that the true ordering of $B$ is consistently estimated. The estimated matrix $\widehat{B}$ minimizing the loss in Equation (5) converges to the true $B$ with the probability going to 1 as $n\to\infty$.*
>
> The conditions in Theorem A align with those commonly imposed in causal structural learning (e.g., Shi et al. (2021)). Our proof follows similar strategies but accounts for the extra penalty term from causal effects. Notice that the explicate forms of causal effects under LSEM are linear combinations of elements of $B$ (see lines 239-249). This implies our new regulation can similarly vanish away as $n$ goes to infinity. We've included the full proof in the revised paper, although omitted here due to character limitations. In addition, this consistency is also empirically verified by the simulation studies.
>
> 3. > The synthetic setup for non-linear DGPs (data-generating process).
>
> **Re:** We greatly value your recommendation and have tested additional non-linear DGPs for the sample size $n=1000$ and the number of nodes $p=20$. As in **new Table 2 of GR**, the proposed method consistently outperforms all baselines, which demonstrates its applicability in handling complex scenarios. In addition, we further conducted additional real data analysis using the benchmark data from Sachs et al. (2005). To validate our method's capacity to find the NSCG and align with Definition 3.2, we designated the protein Akt as the target outcome (see **Figure 2 in GR**). Our method and seven baseline methods were applied and evaluated against the true NSCG associated with the protein Akt. **Table 4 in GR** shows that our method achieves the best performance in finding the NSCG concerning the protein Akt.
>
> 4. > How Definition 3.1 allows to have features/nodes from that have a path towards Y.
>
> **Re:** Thank you for your excellent question. Definition 3.1 focuses on the causal chain starting from outcome $Y$ and traces back to $Y$'s parents and ancestors, as represented by the set $PA_Y(\mathcal{G})$ (defined in line 95). By comparing the joint probabilities of $Y$’s parents/ancestors and $Y$ itself in the full graph $\mathcal{G}_O$ with that in the sub-graph $\mathcal{G}_V$, we can identify the sub-structure (maybe non-unique per sufficiency’s definition) that containing directed edges or path towards $Y$ to achieve the same joint distribution. Example 4.7 illustrates this process, demonstrating how either a graph containing $[X_F,X_B,X_D]$ or a subgraph with $[X_F,X_B]$ can be a sufficient graph. The application of Definition 3.2 further refines this to identify the minimal substructure, i.e., $X_F\to X_B$, as the NCSG for node $X_B$.
>
> 5. > Why does TPR decrease for NSCSL-DE for the S2-S4 on increasing the sample size?
>
> **Re:** This is a indeed keen observation! As we elaborated in lines 325-327, NSCSL based on TE identifies all causal paths towards the outcome, while NSCSL-DE only uncovers direct relationships. Given that the graphs in S2-S4 contain both direct parents and ancestors, NSCSL-DE retrieves only a subset of the true NSCG, leading to a slightly lower TPR. As the sample size grows, this TPR further decreases, converging to the true rate, reflecting the proportion of direct parents among all parents/ancestors.
>
> We extend our heartfelt appreciation for your thoughtful review and comments. We have made every effort to respond to your concerns accurately and have incorporated these explanations into the revised paper. Please do not hesitate to share any additional comments or questions, as we look forward to your further insights and hope to continue improving our work under your expert insights.

---

> > ### Author Response · Authors · 2023-08-15
> > **Eagerly Looking Forward to Feedback on Our Response**
> >
> > We greatly appreciate the time and effort you've invested in reviewing our work and providing insightful and constructive feedback!
> >
> > In response to your valuable suggestions, we have conducted additional experiments on the non-linear case and further clarified our motivation, definition, and simulation results, followed by an additional theory for graph consistency.
> >
> > We earnestly hope that these clarifications and expanded experiments will thoroughly address your concerns. We are eagerly looking forward to your kind feedback!

---

> > > ### Comment · Reviewer_hnpE · 2023-08-16
> > > **Response to the Authors**
> > >
> > > I really thank the authors for addressing the weakness of the paper and for clarifying my doubts. I feel my questions are answered thus I am increasing the score by 1. Please incorporate (new) Thereom A described in the rebuttal above in the main paper.

---

> > > > ### Author Response · Authors · 2023-08-17
> > > > **Thank you note to Reviewer hnpE**
> > > >
> > > > We sincerely thank you for your continued engagement and kind acknowledgment of our efforts to address your insightful comments. We are thrilled that your questions have been answered, and we deeply appreciate the increased score. As per your valuable suggestion, we will incorporate the new Theorem A into Section 5 of the main paper. Once again, thank you for your constructive and encouraging feedback!

---

### Official Review · Reviewer_vBqs · 2023-07-27

**Soundness:** 3 good
**Presentation:** 2 fair
**Contribution:** 2 fair
**Rating:** 6
**Confidence:** 4

**Summary:**

This paper addresses the challenge of discovering causal relationships among variables within a complex graph by proposing the learning of necessary and sufficient causal graphs (NSCGs). Unlike existing methods that consider all variables in the graph, NSCGs exclusively consist of causally relevant variables for a specific outcome of interest, referred to as causal features. The authors introduce the concept of probabilities of causation to assess the importance of features in the causal graph and identify a subgraph relevant to the outcome.

**Strengths:**

1. The problem (i.e., learning a class of necessary and sufficient causal graphs) studied in this paper is interesting;

2. The necessary and sufficient causal structural learning (NSCSL) algorithm offers a new method to learn NSCGs from data;

3. The authors have provided theoretical support for their approach;

4. The experiments were conducted on both synthetic and real data.

**Weaknesses:**

1. This paper does not explicitly discuss the computational complexity of the NSCSL algorithm compared to other existing algorithms. Causal structure learning is often computational expensive, which significantly restricts its practical utility when the real system is complex with hundreds or even thousands of features. Providing a comparison of the computational requirements and runtime of the proposed algorithm could further enhance the assessment of its workload. This becomes particularly important when considering variable selection methods in causal graphs. Without such a comparison, it remains uncertain whether NSCSL would be practically viable, as it might prove to be more computationally expensive than other established methods. Thus, addressing this aspect is paramount to establishing the practical utility and potential applicability of NSCSL in real-world scenarios.

2. The current experimental results do not convincingly demonstrate the effectiveness of the proposed method. Concerning the synthetic data, the authors have not provided a clear explanation of the ER model and the reasons for choosing it over other models such as the Barab´asi–Albert model or scale-free model. It would be more appropriate to utilize different models to generate diverse synthetic data. Additionally, the generated data is limited to only 100 samples or features for the first three scenarios, making it less practically applicable. Moreover, the evaluation's reliance on relatively old baselines (NOTEARS, PC, ICA) is questionable, and it should include state-of-the-art algorithms like DYNOTEARS, DAG-GNN, and GSGES. Regarding the real data, the paper relies solely on one gene data without a ground-truth causal structure. To enhance the validity, benchmark data sets like the real dataset from Sachs et al. (2005) should be incorporated, as it comes with a consensus network that is accepted by the biological community. Finally, there are several parameters in the proposed method. Parameter sensitivity analysis should be included.

K. Sachs, O. Perez, D. Pe’er, D. A. Lauffenburger, and G. P. Nolan. Causal Protein-Signaling Networks Derived from Multiparameter Single-Cell Data. Science, 2005.

3. This paper is not easy to follow due to so many abbreviations and mathematical notations. To enhance the paper's readability, including a notation table and full form of abbreviations on the current page would be beneficial, ensuring that readers can easily understand the content without the need for constant cross-referencing.

**Questions:**

1. What are the computational complexity and runtime of the NSCSL algorithm compared to other existing algorithms?

2. Why choosing ER model, not the other models, to generate the data?

3. Why not consider more recent algorithms as the baselines?

4. The pros and cons of NSCSL vs. variable selection methods in causal graphs or backtracking on causal graphs from the outcomes of interest.

5. The authors should conduct parameter sensitivity analysis in the experiment.

I have read the author’s rebuttal. Some of my concerns have been addressed.

**Limitations:**

As pointed out by the authors, one potential limitation is the assumption of no unmeasured confounders, which may not always hold in real-world scenarios.

---

> ### Author Rebuttal · Authors · 2023-08-09
>
> Thanks for your valuable comments and suggestions! We are encouraged by your acknowledgment of the interesting task of learning a class of NSCG, our novel NSCGL method, and the theoretical support, coupled with experiments on both synthetic and real data. Below, we summarize your questions and comments in quotes and provide our point-by-point responses. Please refer to **the one-page PDF in the general response (GR)** for all additional simulation/real-data results.
>
>
> 1.  > The computational complexity of the NSCSL algorithm.
>
> **Re**: We appreciate this insightful comment. The computational complexity of NSCSL comprises two parts: the cost from causal discovery as $g(n,p)$, and the estimation of causal effects/scores $f(n,p)$, where $n$ is the data sample size and $p$ is the number of nodes. In the linear case, our method learns the features and causal graph through single-step optimization (detailed in Section 5 and Equation (5)), with complexity cubic in the number of nodes, $g(n,p) = \mathcal{O}(p^3)$ following Zheng et al. (2018). Here, the causal effect computation is linear-time and thus is dominated. In the nonlinear case, according to Appendix A.2, the time complexity depends on the base causal discovery method and the number of max iterations $K$, yielding $ \mathcal{O}[K(g(n,p) + f(n,p))]$. Supporting runtime details follow in the next response.
>
> 2. > A comparison of the computational requirements and runtime.
>
> **Re**: Thank you for this excellent comment. We have included the average running time of NSCSL against benchmarks in all simulation settings. **New Tables 1-3 in GR** reveal that NSCSL is as fast as the quickest benchmarks such as PC, LinGAM, and FCI, and significantly faster than others like DAGGNN, GSGES, and CAM. Our method's integration of treatment effects into the optimization adds efficiency and restricts the searching space, making it practical and even beat NOTEARS in computation.
>
> 3. > Diverse synthetic data such as the Barabási–Albert/scale-free model.
>
> **Re**: We really appreciate your great suggestion. We have included additional simulation results based on the scale-free (SF) model, comparing them to the ER model used in the original paper. Comparing the results of new Scenario 5 ($p=50$, $n=1000$) in **Table 1 and 3 in GR**, our method consistently performs the best in finding the NSCG, regardless of the synthetic data models used.
>
> 4. > The generated data is limited to only 100 samples for the first three scenarios.
>
> **Re**: Thank you for this constructive comment. We have added further simulation results, with the number of nodes increased to 50 (as new Scenario 5) and the sample size increased to 1000 and 3000 for Scenarios 4-5. As shown in **Tables 1-3 in GR**, our method excels in these enhanced settings which highlights our method's practical applicability.
>
> 5. > Include state-of-the-art algorithms like DYNOTEARS, DAG-GNN, and GSGES.
>
> **Re**: Thank you for this excellent suggestion. We've expanded the comparison studies to include four additional state-of-the-art methods, including DAG-GNN, GSGES, FCI, and CAM. Since DYNOTEARS targets non-stationary and time-series data, it does not apply to our focus. The new comparisons encompass Scenario 4 ($p=20$, $n=100, 1000 $) and new Scenario 5 ($p=50$, $n=1000,3000$), under varied settings. As displayed in **Tables 1-3 in GR**, our method outperforms all baseline methods.
>
> 6. > Incorporate the real dataset from Sachs et al. (2005).
>
> **Re**: We value this comment and have conducted additional real data analysis using the benchmark data from Sachs et al. (2005). To validate our method's capacity to find the NSCG and align with Definition 3.2, we designated the protein Akt as the target outcome. This designation ensures that NSCG exists (see **Figure 2 in GR**) and that finding an NSCG is meaningful. Our method and seven baseline methods were applied and evaluated against the true NSCG associated with the protein Akt. **Table 4 in GR** shows that our method achieves the best performance in finding the NSCG concerning the protein Akt.
>
> 7. > Parameter sensitivity analysis should be included.
>
> **Re**: Thank you for your excellent suggestion. We have conducted comprehensive sensitivity analyses concerning all hyperparameters listed in Table D.1 in the appendix. This includes the L1 penalty, the maximum ascent steps, the tolerance level, and the upper limit of the dual updating. These results, evaluated using Scenario 4 ($p=20$, $n=1000$), are presented in **Figure 1 in GR**, indicating that our method remains robust to these parameters, provided they are set within a reasonable range.
>
> 8. > Abbreviations and mathematical notations.
>
> **Re**: We appreciate this constructive feedback and have compiled a notation table and the full forms of abbreviations mentioned throughout the paper. This information is now included in the revised paper.
>
> 9. > The pros and cons of NSCSL vs variable selection methods.
>
> **Re**: This insightful comment prompted us to clarify the distinctions between our proposed method and existing variable selection methods. The primary advantage of NSCSL is that it concurrently learns the causal graph while selecting the causal features. In contrast, existing methods, including variable selection in causal graphs or backtracking on causal graphs, depend on a true or known graph to conduct feature selection for the outcome or target of interest. While our approach introduces additional time costs to learn the unknown causal graph, along with potential estimation errors, its holistic approach offers significant benefits in uncovering the causal features.
>
> We extend our sincere gratitude to your thorough examination of our paper. These insights and suggestions have substantially enriched our work. All the above clarifications, discussions, and improvements are now part of the revised manuscript. We are eager to address any further comments or suggestions and look forward to your continued feedback.

---

> > ### Author Response · Authors · 2023-08-15
> > **Eagerly Looking Forward to Feedback on Our Response**
> >
> > We deeply appreciate the time and effort you have devoted to reviewing our work and providing us with insightful, detailed, and encouraging feedback.
> >
> > Following your constructive suggestions, we have conducted six additional sets of experiments and have further clarified our computational complexity and advancements. All of these details can be found in our response and within the one-page PDF file.
> >
> > We sincerely hope our further clarifications and experiments can fully address your concerns. We are eagerly looking forward to your kind feedback!

---

> > > ### Comment · Reviewer_vBqs · 2023-08-16
> > >
> > > I really appreciate the authors for the detailed response and additional experimental results. Some of my concerns have been addressed. However, my primary concern still revolves around the computational complexity of the proposed algorithm. While the authors acknowledge in their rebuttal that their approach would "incur additional time costs," this aspect raises reservations in my mind regarding the practical utility of their research. In my view, the true value of a variable selection method lies in its ability to streamline a set of promising features/variables for the purpose of accelerating causal structure learning.
> > >
> > > Furthermore, I would like to pose an additional question regarding the running time in Tables 1-3 of GR. These tables suggest that the PC algorithm is among the fastest algorithms. Nonetheless, drawing from my own experience, I find that the PC algorithm tends to be very computationally expensive (i.e., can not be applied to real scenarios at all). Could you specify the variant of the PC algorithm that you employed in your work? Thanks.

---

> > > > ### Author Response · Authors · 2023-08-17
> > > > **Response to Official Comment by Reviewer vBqs**
> > > >
> > > > We are very grateful for your continued feedback and acknowledgment of our detailed response and newly-conducted experiments. To address your additional comments on computational complexity and the PC algorithm, we'd like to further clarify the following aspects.
> > > >
> > > > 1. **Faster computational time compared to the base learner**: It is essential to highlight that by incorporating causal effects as an additional regulator into the optimization objective in equation (5), our method restricts searching in a smaller space, enhancing learning efficiency and speed, and thus can even slightly faster than NOTEARS (our base learner in experiments) in computation time. In addition, our proposed NSCSL is on par with some of the fastest benchmarks such as PC, LinGAM, and FCI, and considerably quicker than others like DAGGNN, GSGES, and CAM, as shown in Tables 1-3 in GR. Thus, *our algorithm not only is practical but also accelerates score-based causal structure learning methods*.
> > > >
> > > > 2. **Clarification of the phrase "incur additional time costs"**: We wish to clarify that the “additional time costs” arise when comparing our method to variable selection techniques that rely on already known causal graphs/relationships. In such scenarios, the extra time pertains to learning the unknown causal graph. Yet, our proposed method is flexible to be extended to learn NSCG when the causal graph is known, where no additional time cost will be paid. Hence, *our method is more practically useful as it applies whether the causal graph is known or unknown*.
> > > >
> > > > 3. **Implementation of the PC algorithm**: We employ the Peter-Clark (PC) function from the ***causal-learn*** Python package [1], using default settings including Fisher’s Z conditional independence test, with a desired significance level of 0.05.
> > > >
> > > > [1] Zheng, Y., Huang, B., Chen, W., Ramsey, J., Gong, M., Cai, R., Shimizu, S., Spirtes, P. and Zhang, K., 2023. Causal-learn: Causal Discovery in Python. arXiv preprint arXiv:2307.16405.
> > > >
> > > > 4. **The computational time of the PC algorithm**: We concur with your assessment of the PC algorithm. While it ranks among the fastest algorithms in Tables 1-3 of GR (for the number of nodes as $p=20,50$), the PC algorithm becomes extremely slow when applying to the real yeast gene data (where ***$p=492$***), *exceeding a 48-hour run time*. In contrast, our method and most other baselines completed the real data task within 6 hours.
> > > >
> > > > We sincerely hope that these clarifications satisfactorily address your comments. We remain devoted to refining our paper and welcome any further questions or insights. Your expertise and thoughtful suggestions are deeply appreciated, and we thank you once again for your vital contributions to our work!

---

> > > > > ### Comment · Reviewer_vBqs · 2023-08-19
> > > > >
> > > > > Thank you for the explanation! Although I still have some concern on the practical value of the proposed method, I have increased my score by 1 since the authors addressed most of my other concerns.

---

> > > > > > ### Author Response · Authors · 2023-08-21
> > > > > > **Thank you note to Reviewer vBqs**
> > > > > >
> > > > > > We sincerely thank you for your continued engagement and kind acknowledgment of our explanation. We are thrilled that most of your concerns have been properly addressed, and we deeply appreciate the increased score. We've included the related discussions in our paper to enhance the practical value of the proposed method. Once again, thank you for your constructive and encouraging feedback!

---

### Author Rebuttal · Authors · 2023-08-10

We extend our heartfelt thanks to all reviewers for their insightful comments and suggestions. We are encouraged by their highlight of **various acknowledgments**, which affirm the quality and novelty of our work, as summarized below:

- Reviewer vBqs appreciated the novel NSCGL method and its theoretical backing and commended the experiments on synthetic and real data.
- Reviewer hnpE lauded the paper for exploring less-trodden grounds in variable selection for causal graph learning, noting the novelty of Theorem 4.6 and the method's consistent improvement over other baselines. The clear writing style was also praised.
- Reviewer LXJs recognized the well-written paper, illustrative examples, and open discussion on limitations, finding it interesting to read.
- Reviewer Qe9n approved the work as an essential and well-defined problem, appreciated the intuitive NSCG, and found the extension of the POC concept intriguing.
- Reviewer hThF commended the motivation, introduction, comprehensive definitions, and explanations, recognizing the relevance of the method across various causal inference problems.

### **Summary of Common Comments and Added Simulation/Real-data Results**

Next, we summarize the **common questions/comments raised by the reviewers** in quotes and then provide our point-by-point responses. Please refer to **the one-page PDF in the general response (GR)** for all additional simulation/real-data results we conducted.

1. > The generated data is limited to only 100 samples for the first three scenarios (Reviewers vBqs, LXJs, Qe9n)

**Re**: We have added further simulation results, with the number of nodes increased to 50 (as new Scenario 5) and the sample size increased to 1000 and 3000 for Scenarios 4-5. As shown in **Tables 1-3 in GR**, our method excels in these enhanced settings which highlights our method's practical applicability.

2.  > Include more and new state-of-the-art algorithms (Reviewers vBqs, LXJs, hThF)

**Re**: We've expanded the comparison studies to include four additional state-of-the-art methods, including DAG-GNN (suggested by reviewer vBqs), GES with generalized score (GSGES, suggested by reviewer vBqs), FCI (suggested by reviewer LXJs), and CAM (a generalized version of LinGAM). The new comparisons encompass Scenario 4 ($p=20$, $n=100,1000$) and new Scenario 5 ($p=50$, $n=1000,3000$), under varied settings. As displayed in **Tables 1-3 in GR**, our method outperforms all baseline methods.

3. > Incorporate the real dataset from Sachs et al. (2005) (Reviewers vBqs, LXJs, Qe9n)

**Re**: We have conducted additional real data analysis using the benchmark data from Sachs et al. (2005). To validate our method's capacity to find the NSCG and align with Definition 3.2, we designated the protein Akt as the target outcome. This designation ensures that NSCG exists (see **Figure 2 in GR**) and that finding an NSCG is meaningful. Our method and seven baseline methods were applied and evaluated against the true NSCG associated with the protein Akt. **Table 4 in GR** shows that our method achieves the best performance in finding the NSCG concerning the protein Akt.

4. > The synthetic setup for non-linear DGPs (data-generating process) (Reviewers hnpE, hThF)

**Re:** We have tested additional non-linear DGPs for the sample size $n=1000$ and the number of nodes $p=20$. As in **new Table 2 of GR**, the proposed method consistently outperforms all baselines, which demonstrates its applicability in handling complex scenarios.

### **Additional Notable Suggestions and More Simulation/Real-data Results**

Besides the common comments we received, we would like to further summarize **other additional simulation/real-data results we provided** to adjust many notable and valuable suggestions from the reviewers.

5. > A comparison of the computational requirements and runtime (Reviewer vBqs)

**Re**: We have included the average running time of NSCSL against benchmarks in all simulation settings. **New Tables 1-3 in GR** reveal that NSCSL is as fast as the quickest benchmarks such as PC, LinGAM, and FCI, and significantly faster than others like DAGGNN, GSGES, and CAM. Our method's integration of treatment effects into the optimization adds efficiency and restricts the searching space, making it practical and even beating NOTEARS in computation.

6. > Diverse synthetic data such as the Barabási–Albert/scale-free model (Reviewer vBqs)

**Re**: We have included additional simulation results based on the scale-free (SF) model, comparing them to the ER model used in the original paper. Comparing the results of new Scenario 5 ($p=50$, $n=1000$) in **Table 1 and 3 in GR**, our method consistently performs the best in finding the NSCG, regardless of the synthetic data models used.

7. > Parameter sensitivity analysis should be included (Reviewer vBqs)

**Re**: We have conducted comprehensive sensitivity analyses concerning all hyperparameters listed in Table D.1 in the appendix. This includes the L1 penalty, the maximum ascent steps, the tolerance level, and the upper limit of the dual updating. These results, evaluated using Scenario 4 ($p=20$, $n=1000$), are presented in **Figure 1 in GR**, indicating that our method remains robust to these parameters, provided they are set within a reasonable range.

8. > Real data compare to NOTEARS only (Reviewer LXJs)

**Re:** We have conducted additional real data analysis using all baseline methods. The summarized results in **Table 5 in GR** highlight our method's ability to identify relevant genetic influencers for the variant YER124C without contamination by irrelevant genes.

We extend our sincere gratitude to all reviewers' thorough examination of our paper. These insights and suggestions have substantially enriched our work. All the above clarifications, discussions, and improvements are now part of the revised manuscript. We are eager to address any further comments or suggestions and look forward to all your continued feedback.

---

### Decision · Program_Chairs · 2023-09-21

**Decision:**

Accept (spotlight)

**Comment:**

The paper bridges two streams of work in causality: general full graph discovery algorithms and outcome-targeted algorithms that do not need the full graph. As reviewers noted, the paper makes a convincing case for detecting necessary and sufficienct variables for an outcome and then doing discovery only on those outcomes. Reviewers appreciated the theoretical contributions and the authors have extended their empirical experiments in the rebuttal. I encourage the authors to update their paper based on the feedback.